# SYNQUE: ESTIMATING SYNTHETIC DATASET QUALITY WITHOUT ANNOTATIONS

## ABSTRACT

We introduce and formalize the **Synthetic Dataset Quality Estimation** (SYNQUE) problem: ranking synthetic datasets by their expected real-world task performance using only limited unannotated real data. This addresses a critical and open challenge where data is scarce due to collection costs or privacy constraints. We establish the first comprehensive benchmarks for this problem by introducing and evaluating proxy metrics that choose synthetic data for training to maximize task performance on real data. We introduce the first proxy metrics for SYNQUE by adapting distribution and diversity-based distance measures to our context via embedding models. To address the shortcomings of these metrics on complex planning tasks, we propose LENS, a novel proxy that leverages large language model reasoning. Our results show that SYNQUE proxies correlate with real task performance across diverse tasks, including sentiment analysis, Text2SQL, web navigation, and image classification, with LENS consistently outperforming others on complex tasks by capturing nuanced characteristics. For instance, on text-to-SQL parsing, training on the top-3 synthetic datasets selected via SYNQUE proxies can raise accuracy from 30.4% to 38.4 (+8.1)% on average compared to selecting data indiscriminately. This work establishes SYNQUE as a practical framework for synthetic data selection under real-data scarcity and motivates future research on foundation model-based data characterization and fine-grained data selection.

## 1 INTRODUCTION

Data scarcity hinders effective machine learning, especially for tasks requiring specialized expertise like autonomous navigation or natural language interfaces, where data collection is costly and slow (Xie et al., 2024; Yang et al., 2024). In sensitive domains such as healthcare and finance (Tan et al., 2024; Jordan & Mitchell, 2015), privacy concerns further complicate data acquisition. Large generative models have emerged as capable synthetic data generators, producing annotated data for tasks like policy learning (Xu et al., 2024), Text2SQL (Yang et al., 2024), sentiment analysis (Ye et al., 2022; Li et al., 2023c), and image classification (Geng et al., 2025). While synthetic data can improve real-world performance under scarcity, results vary widely depending on task and data quality (Huang et al., 2025; Geng et al., 2025).

Can we distinguish between high-quality synthetic data that improves real-world task performance and low-quality data that offers little benefit, *without any annotated real data* and *without costly model training*? Crucially, increasing the size of synthetic datasets does not always lead to better downstream performance as it does with real data; in some cases, larger synthetic datasets can even degrade performance, exhibiting inverse scaling trends (Geng et al., 2025; Li et al., 2023c; Setlur et al., 2024; Gao et al., 2022; Møller et al., 2023). Therefore, selecting a synthetic dataset from a pool of datasets to train on to optimize downstream performance is important.

We introduce **Syn**thetic Dataset **Qu**ality **E**stimation, or SYNQUE, the problem of ranking multiple synthetic datasets by quality using only limited unannotated samples of real data. A synthetic dataset A is of higher quality than B if a model trained on A outperforms one trained on B on a real-world test set. This ability is crucial when real data annotation is costly or infeasible. For example, in text-to-SQL parsing, SYNQUE helps select the synthetic dataset that yields better generalization from a small set of unannotated real queries. Similarly, for intelligent web agents, it identifies the synthetic interactions that produce agents performing best on real navigation tasks.

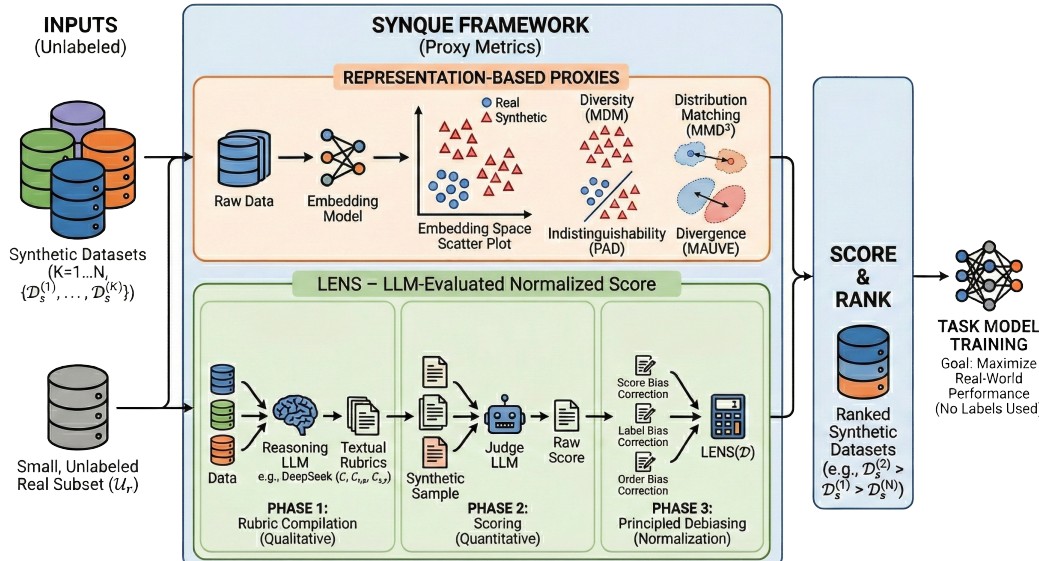

Figure 1: SYNQUE uses synthetic data and unlabeled samples of real data to estimate synthetic data quality. Proxy scores are used to rank and select the datasets that lead to the best task performance.

This work makes three main contributions. 1) We formalize the SYNQUE problem and establish the first comprehensive benchmark. As part of this, we introduce and evaluate a suite of *proxy metrics*: computable scores that estimate a synthetic dataset's quality using only a subset of unannotated real data. We adapt proxy metrics using established distributional measures such as mean distance to medoids (Cox et al., 2021), diversity measures such as Proxy-A-Distance (Ben-David et al., 2006), and divergence measures such as MAUVE (Pillutla et al., 2021) — none have been systematically evaluated for the purpose of synthetic data selection. 2) We propose **LLM-E**valuated **N**ormalized **S**core (LENS), a novel proxy measure that leverages large language model (LLM) reasoning to create *dataset rubrics* that highlight difference between synthetic data and real data in language. 3) We conduct a comprehensive experiment across diverse domains in sentiment analysis, Text2SQL, web navigation, and image classification in order to evaluate how well these proxy metrics are able to select synthetic data to maximize performance on real test data.

Our empirical evaluation shows that SYNQUE proxies exhibit moderate to strong correlation with real task performance across diverse domains including sentiment analysis, Text-to-SQL, image classification, and web navigation. While the best proxy varies by task, most can effectively predict downstream performance without *any* labeled real data, enabling practical synthetic data selection that outperforms indiscriminate synthetic data selection. We find that the reliability of proxies depends on task complexity, with higher variance on noisy data like synthetic images. Among the proxies, LENS, leveraging LLM reasoning and a principled debiasing strategy, consistently achieves superior gains on complex tasks like web navigation by capturing nuanced task details beyond embedding-based metrics. These results establish SYNQUE as a robust framework for selecting high-quality synthetic data and motivate future work on stronger foundation model methods to characterize data and perform fine-grained, example-level data selection.

## 2 RELATED WORK

**Data synthesis**  Synthesizing training data with generative models is a promising way to address data scarcity by leveraging instruction-following abilities (Touvron et al., 2023; Ouyang et al., 2022) and vast pre-trained knowledge (Long et al., 2024; Honovich et al., 2022; Mishra et al., 2022). This has been applied across domains such as text classification (Ye et al., 2022), Text-to-SQL (Yang et al., 2024; Lei et al., 2024; Li et al., 2024), planning (Sun et al., 2024; Hu et al., 2024; Xu et al., 2024; Murty et al., 2025), and computer vision (Geng et al., 2025; Li et al., 2022). While synthetic data often improves downstream models (Liu et al., 2024; Ye et al., 2022), challenges remain in ensuring

quality due to issues such as hallucination (Huang et al., 2025), mode collapse (Goodfellow et al., 2014; Durall et al., 2020; Shumailov et al., 2024), and counterfactual artifacts (Li et al., 2023c; Yu et al., 2023). Our work introduces the SYNQUE framework for selecting synthetic data to maximize task performance without access to large amounts of real data and without training task models.

**Distributional and diversity metrics** Metrics like Proxy-$\mathcal{A}$-Distance (**PAD**) estimate domain divergence by training classifiers to distinguish source and target data (Ben-David et al., 2010; 2006; Quinonero-Candela et al., 2022). While effective in some NLP tasks (Elsahar & Gallé, 2019), **PAD** struggles in noisy, complex settings such as agent planning (He et al., 2024). Diversity metrics like mean-distance-to-medoids (**MDM**)(Cox et al., 2021) and DCScore(Zhu et al., 2025) approximate the coverage of synthetic data. However, these measures do not capture common synthetic data factual inconsistencies from generative model hallucinations (Huang et al., 2025; Li et al., 2023b; Gunjal et al., 2023), which can harm downstream performance (Geng et al., 2025; Yu et al., 2023; Casco-Rodriguez et al., 2023; Li et al., 2023c; Hataya et al., 2023; Briesch et al., 2023).

**Data selection and weighting** A growing body of work focuses on selecting high-value training samples. Some approaches utilize scaling laws by training regression models on short runs to predict the best data mixture (Liu et al., 2025; Magnusson et al., 2025). Others employ influence functions or gradient-based heuristics to select influential subsets (Zhang et al., 2024; Hu et al., 2025), or utilize small real datasets to learn instance-level weights for synthetic data (Kuo et al., 2025). Crucially, these methods focus on *instance-level* selection or weighting, often requiring access to labeled validation sets to compute gradients or train selection models. In contrast, SYNQUE addresses *dataset-level* ranking in a zero-shot, gradient-free manner, making it suitable for "cold-start" settings where no labeled data exists. Finally, another group of works try to distinguish between human and machine text using divergence measures (Pillutla et al., 2021) or LLM-as-a-judge (Gu et al., 2025; Krumdick et al., 2025; Zheng et al., 2023). Neither of these two techniques have been studied for synthetic data selection. Furthermore, the latter also suffers from limitations of generative models previously mentioned such as hallucination. Our work establishes the first systematic benchmark and results for synthetic data selection in a practical, fully unsupervised regime.

# 3 THE SYNTHETIC DATASET QUALITY ESTIMATION PROBLEM

We define the SYNQUE problem and establish notation. Let $\mathcal{D}_r = \{x_r^{(i)}, y_r^{(i)}\}_{i=1}^{n_r}$ be a real dataset, which we assume is scarce and for which labels are unavailable for validation. Instead, we only have access to a small, unannotated collection of real-world inputs $\mathcal{U}_r = \{x_i\}_{i=1}^{m_r} \in \mathcal{D}_r$. Our goal is to use $\mathcal{U}_r$ to estimate the quality of $K$ synthetic datasets $\{\mathcal{D}_s^{(1)}, \ldots, \mathcal{D}_s^{(K)}\}$. These datasets might be generated by different methods and thus vary in quality. We aim to select the synthetic dataset $\mathcal{D}_s^*$ that yields the best model performance on $\mathcal{D}_r$, *without using labeled real data or training models on any synthetic dataset*.

Consider a model $f(\cdot; \theta_k)$ trained on the synthetic dataset $\mathcal{D}_s^{(k)}$ with parameters $\theta_k$. Let $M(f, \mathcal{D}_e)$ denote model $f$'s performance on the evaluation dataset $\mathcal{D}_e$, for instance task completion rate or accuracy. The ideal synthetic dataset $\mathcal{D}_s^*$ satisfies:

$$\mathcal{D}_s^* = \operatorname{argmax}_{\mathcal{D}_s^{(k)}} M\left(f\left(\cdot; \theta_k\right), \mathcal{D}_e\right) \tag{1}$$

In practice, Eq 1 is infeasible because it requires labeled real data for evaluation and extensive model training across $K$ synthetic datasets. Instead, SYNQUE seeks *proxy metrics* $Q(\mathcal{D}_s^{(k)}, \mathcal{U}_r)$—computable using only the synthetic dataset $\mathcal{D}_s^{(k)}$ and unannotated real samples $\mathcal{U}_r$—that correlate strongly with true downstream performance. The SYNQUE problem thus reduces to designing or learning a proxy function $Q : (\mathcal{D}_s^{(k)}, \mathcal{U}_r) \to \mathbb{R}$ such that a higher score indicates the synthetic dataset $\mathcal{D}_s^{(k)}$ is more likely to produce better real-world task performance. This formulation enables efficient and effective use of synthetic data in low-resource or privacy-sensitive scenarios. For ease of exposition, we refer to the score produced by the proxy function $Q$ as the **SYNQUE score**.

| Listing 1: Rubric prompt | Listing 2: Scorer Prompt |
|---|---|
| ```
You are shown samples
from datasets A and B.
Give up to 10 points
describing how dataset B
is similar to dataset A.

Samples from dataset A:
${location A}

Samples from dataset B:
${location B}
``` | ```
Given similarities and differences between
datasets, how likely is the given sample
from dataset ${prediction}? Choose from very
unlikely, unlikely, unsure, likely, and very
likely.

Similarities: ${similarities}

Differences: ${differences}

Sample: ${sample}
``` |

Table 1: Simplified LENS prompt templates. Appendix C contains detailed prompts for each task.

## 4 SYNQUE PROXY METRICS

In this section, we introduce the suite of proxy metrics designed to solve the SYNQUE problem. We study the potential of leveraging several distributional distance measures as a vehicle to quantify synthetic datasets quality. As the distributional distance measures have not been applied to synthetic dataset evaluation nor for tackling SYNQUE, we adapt them to tackle SYNQUE by embedding raw text or image data into representations, so distributional distances can be quantified through these traditional measures.

To address representation-based failure on evaluating long-horizon tasks, we introduce LENS a LLM-based measure that operates directly on the raw data to be evaluated, therefore providing contextual understanding instead of sole depending on a universal embedding model. We introduce both our adapted representation-based metrics and LLM-based metrics in the following section.

### 4.1 REPRESENTATION-BASED PROXY METRICS

**Mean Distance to Medoid** Our first proxy metric adapts Mean Distance to Medoid (**MDM**), a measure of dataset diversity (Cox et al., 2021; Laliberté & Legendre, 2010; Lehman & Stanley, 2011; Risi et al., 2009). The rationale is that high-diversity synthetic datasets may offer broader coverage in the embedding space and thus yield higher performance on real data. For SYNQUE, **MDM** characterizes this diversity by measuring the sparsity of data points around medoids. Given $\mathcal{D}_\mathrm{s}$, we compute $N$ medoids using clustering algorithms such as kMedoids [1]. For each medoid $\tilde{x}^M$, we aggregate the Euclidean distances from all points within their corresponding cluster: $\mathbf{MDM} = \frac{1}{N} \sum_{i=1}^{N} d(x_i^M, \tilde{x}^M)$. Intuitively, if points within each cluster are centered around the medoid, **MDM** will be small, and therefore less diverse in the embedding space. In contrast, high-diversity synthetic datasets to have broader coverage, and therefore higher performance on real data. A higher **MDM** score suggests greater diversity, so we use it directly as the SYNQUE score.

**Maximum Mean Discrepancy** Next, we propose a proxy based on Maximum Mean Discrepancy ($\mathbf{MMD}^2$), a nonparametric test that assesses whether two samples originate from the same distribution (Gretton et al., 2012; Borgwardt et al., 2006; Lu et al., 2022; Li et al., 2015). As a SYNQUE proxy, we use $\mathbf{MMD}^2$ to measure the discrepancy between the synthetic dataset and the real data distribution. Given $n_\mathrm{s}$ synthetic input samples $x_i \in \mathcal{D}_\mathrm{s}$ of size $n_\mathrm{s}$ and $m_\mathrm{r}$ unannotated real input samples $y_i \in \mathcal{U}_\mathrm{r}$, $\mathbf{MMD}^2$ quantifies the distance between these two empirical distributions in a reproducing kernel Hilbert space using a kernel function $k(\cdot)$.

$$\mathbf{MMD}^2 = \frac{1}{m_\mathrm{r}^2} \sum_{i,j=1}^{m_\mathrm{r}} k(x_i, x_j) + \frac{1}{n_\mathrm{s}^2} \sum_{i,j=1}^{n_\mathrm{s}} k(y_i, y_j) - \frac{2}{m_\mathrm{r} n_\mathrm{s}} \sum_{i,j=1}^{n_\mathrm{s}, m_\mathrm{r}} k(x_i, y_j) \quad (2)$$

A smaller $\mathbf{MMD}^2$ score indicates the synthetic data distribution is closer to the real one. To maintain consistency with other proxies where higher is better, we use $-\mathbf{MMD}^2$ as the SYNQUE score.

---

[1] https://pypi.org/project/kmedoids/

**Proxy-A-Distance**   Our third representation-based proxy adapts Proxy-A-Distance (**PAD**), a discriminative measure from domain adaptation that quantifies the divergence between two distributions (Ben-David et al., 2006; Elsahar & Gallé, 2019). The **PAD** proxy measures how well a classifier can discriminate between samples from the synthetic dataset $\mathcal{D}_\mathrm{s}$ and real dataset $\mathcal{D}_\mathrm{r}$. We compute $\mathbf{PAD} = 1 - 2\mathcal{E}(G)$ by training a binary domain classifier $G : x \to [0, 1]$ (e.g., a linear SVM, a multi-layer perceptron) to distinguish between synthetic and real inputs (i.e. we do not assume access to labels), where the error of the classifier $\mathcal{E}$ can be computed as:

$$\mathcal{E}(G) = 1 - \frac{1}{n_\mathrm{s} + m_\mathrm{r}} \sum_{x_i \in \mathcal{D}_\mathrm{s}, \mathcal{U}_\mathrm{r}} |G(x_i) - \mathbb{I}(x_i \in \mathcal{D}_\mathrm{s})| \tag{3}$$

A higher classification error implies the datasets are not easily separable, indicating lower divergence and thus higher synthetic data quality. For consistency with other divergence metrics, use $-\mathbf{PAD}$ as the SYNQUE score.

**MAUVE**   Our final representation-based proxy is an adaptation of MAUVE (Pillutla et al., 2021), a metric that quantifies the divergence between text distributions. MAUVE summarizes both Type I and Type II errors. Given a synthetic data distribution Q and a real data distribution P, Type I error means Q generates text that is unlikely under P (unrealistic samples), while Type II error means Q fails to generate text that is plausible under P (lacks diversity). MAUVE captures both error types in a single score, which approaches 1 as the distributions become more similar. Since a higher score indicates better alignment with real data, we use the MAUVE score directly as the SYNQUE score.

## 4.2   LLM-EVALUATED NORMALIZED SCORE (LENS)

The representation-based proxies described so far rely on high-quality continuous representations of inputs. In low-resource settings where such representations are unavailable, or in long-horizon settings where it is intractable to compress a long sequence of observations and states into a compact, fixed-size representation, these representation-based proxies may fall short. To address this, we introduce **LLM-E**valuated **N**ormalized **S**core (LENS), a novel method that leverages LLMs as zero-shot discriminators. LENS first derives a language **rubric** describing the similarities and differences between samples of unannotated real data $\mathcal{U}_\mathrm{r}$ and inputs from the synthetic dataset $\mathcal{D}_\mathrm{s}$. A subsequent (potentially smaller) LLM then scores how likely each synthetic example is to belong to the real dataset, guided by the rubric. The average score across the synthetic dataset is used as the final SYNQUE score. The intuition is similar to that behind **PAD**: a higher classification error by the rubric-guided scorer implies higher synthetic data quality. We now detail how LENS is computed.

**Rubric compilation**   Given real input samples $\mathcal{U}_\mathrm{r}$, we collect an equal number of samples $\mathcal{U}_\mathrm{s}$ from the synthetic dataset $\mathcal{D}_\mathrm{s}$. Both collections are given to a reasoning LLM (e.g. `DeepSeek R1` or `o4-mini`) to generate three sets of **characteristic descriptions**: commonalities ($C$), differences of real from synthetic ($C_\mathrm{r,s}$), and differences of synthetic from real ($C_\mathrm{s,r}$). Listing 1 shows a simplified rubric compilation prompt template. Our design is backed by the principled idea of approximating domain divergence through discriminator error (Ben-David et al., 2006): LENS's scoring is motivated by **PAD**, where the error of a classifier (here, the LLM-based scorer) reflects the distance between distributions. Unlike **PAD**, however, LENS does not require a pretrained encoder to map samples into fixed-length representations; instead, it operates directly on the native data format and characterizes differences using language rubrics.

**Principled Debiasing and Scoring**   We now describe how to compute the score of a synthetic dataset. A key challenge of scoring is in mitigating LLM biases. We identified three primary sources:

1. **Order Bias:** The set of differences an LLM derives when comparing A to B can differ significantly from when comparing B to A.

2. **Label Bias:** When asked how likely an example $x$ belongs to A or B, an LLM may score both as "very likely", a contradiction.

3. **Score Bias:** LLMs may have an inherent preference for certain score values (e.g., "likely") regardless of the input.

To address these systematically, we employ a minimal design involving four scoring permutations for each sample. We denote the LLM scoring function as $g_{\mathcal{D}|C}$, which outputs a score (0-4) for how likely an example $x$ belongs to dataset $\mathcal{D}$ given characteristic descriptions $C$.

First, to mitigate **score bias**, we compute baseline scores by averaging the LLM's judgments on real inputs $x \in \mathcal{U}_{\mathrm{r}}$ for each of the four permutations. For instance, the baseline for scoring an example as real, given the description of how synthetic differs from real, is:

$$z_{\mathrm{r}|C_{\mathrm{s,r}}} = \mathbb{E}\left[g_{\mathrm{r}|C_{\mathrm{s,r}}}(x)\right] \approx \frac{1}{n_{\mathrm{r}}} \sum_{i=1}^{n_{\mathrm{r}}} g_{\mathrm{r}|C_{\mathrm{s,r}}}(x_i) \tag{4}$$

We then compute a *score-debiased* score $h$ for each synthetic sample by normalizing its raw score against this baseline:

$$h_{\mathrm{r}|C_{\mathrm{s,r}}} = \frac{g_{\mathrm{r}|C_{\mathrm{s,r}}}(x)}{\max(\epsilon, z_{\mathrm{r}|C_{\mathrm{s,r}}})} \tag{5}$$

Here, $\epsilon$ is a small constant to avoid division by zero. Intuitively, this scores-debiased score expresses how much the LLM scores the example compared to how it usually scores real examples. Next, to compute a **label-debiased** score, we normalize the LLM's preference for the "real" label over the "synthetic" label for each synthetic example:

$$p_{\mathrm{r}|C_{\mathrm{s,r}}} = \frac{h_{\mathrm{r}|C_{\mathrm{s,r}}}}{h_{\mathrm{r}|C_{\mathrm{s,r}}} + h_{\mathrm{s}|C_{\mathrm{s,r}}} + \epsilon} \tag{6}$$

Finally, to create an **order-debiased** score, we average the label-debiased scores obtained using both sets of difference descriptions ($C_{\mathrm{s,r}}$ and $C_{\mathrm{r,s}}$):

$$\hat{p}(x) = \frac{1}{2}\left[p_{\mathrm{r}|C_{\mathrm{s,r}}}(x) + p_{\mathrm{r}|C_{\mathrm{r,s}}}(x)\right] \tag{7}$$

The final LENS score for a synthetic dataset is the empirical mean of these fully debiased scores across all its samples:

$$\mathrm{LENS}(\mathcal{D}_{\mathrm{s}}) = \mathbb{E}\left[\hat{p}(x)\right] = \frac{1}{n} \sum_{i=1}^{n_{\mathrm{s}}} \hat{p}(x_i) \tag{8}$$

## 5 EXPERIMENTS

We choose four diverse tasks spanning different machine learning domains to examine how well each candidate proxy metric extrapolate to real data performance on tasks with varying complexities and modalities. For LENS, we incorporate `Deepseek-R1` to generate 10 points about similar and different characteristics $C_{\mathrm{s,r}}$ between synthetic data samples $\mathcal{U}_{\mathrm{s}}$ and real data samples $\mathcal{U}_{\mathrm{r}}$. We then use `Qwen2.5-32B-Instruct` (Qwen et al., 2025) and `8B` to score synthetic examples according to the rubric. For image domain, we use OpenAI `o4-mini` to compile rubrics and `Qwen2.5-VL-32B-Instruct` to score. For representation-based metrics **PAD**, **MDM**, and **MMD**$^2$, we use state-of-the-art `qte-Qwen2-7B-Instruct` to embed text inputs and `E5-V` (Jiang et al., 2024) to embed image inputs for proxy scoring. We use XGBoost (Chen & Guestrin, 2016) to compute **PAD** and polynomial kernel for **MMD**$^2$ (Gretton et al., 2012). We include additional kernel ablations in Appendix 11. We use the official release[2] from MAUVE, with the default hyperparameter setting for MAUVE calculation.

We also include an experiment with perplexity-based metric as additional baseline in Text2SQL (see table 5). PERPLEXITY, inspired by scaling law methods (Magnusson et al., 2025; Liu et al., 2025), fine-tunes a model on each synthetic data set and measures its perplexity on the unannotated real data subset; a lower perplexity is expected to indicate higher quality.

We use Pearson (Pearson & Galton, 1997) and Spearman rank (Spearman, 1904) correlation coefficients to measure how strongly task performance and proxy scores are related. Pearson focuses on *predictability* by capturing linear relationships, while Spearman focuses on *trend* by evaluating whether the relationship between variables is consistently increasing or decreasing (i.e., monotonic), regardless of the exact shape. To reduce variance in correlation analysis across different sample subsets, for all tasks, we construct subsets $\mathcal{U}_{\mathrm{r}}$ by sampling with five different seeds. Final correlation scores are averaged across seeds.

---

[2] https://pypi.org/project/mauve-text/

Table 2: Top-3 task performance for all candidate proxies of SYNQUE. Top-3 task performance is computed by averaging task performance of synthetic datasets chosen using each proxy metric. Improvements are calculated based on increase over average performance of all synthetic datasets.

| Tasks | Test mean | LENS 7B | | LENS 32B | | PAD | MMD$^2$ | MDM | Mauve |
|---|---|---|---|---|---|---|---|---|---|
| | | debiased | biased | debiased | biased | | | | |
| **Sentiment** | 49.6 | 50.5 +0.8 | 51.2 +1.6 | 52.0 +2.4 | 51.0 +1.4 | 55.3 +5.7 | 54.7 +5.1 | 54.2 +4.6 | 54.6 +4.9 |
| **Text2SQL** | | | | | | | | | |
| Computer | 45.8 | 46.6 +0.7 | 46.4 +0.6 | 48.3 +2.5 | 46.3 +0.5 | 48.3 +2.5 | 47.4 +1.6 | 48.2 +2.3 | 48.3 +2.5 |
| Apps | 30.4 | 34.7 +4.4 | 36.3 +5.9 | 33.8 +3.4 | 35.2 +4.9 | 33.5 +3.2 | 38.4 +8.1 | 33.9 +3.5 | 38.4 +8.1 |
| Movies | 37.3 | 41.0 +3.7 | 41.4 +4.1 | 43.8 +6.5 | 44.7 +7.4 | 44.2 +6.9 | 43.0 +5.7 | 46.9 +9.6 | 44.6 +7.3 |
| *Average* | 37.8 | 40.8 +2.9 | 41.4 +3.5 | 42.0 +4.1 | 42.1 +4.3 | 42.0 +4.2 | 42.9 +5.1 | 43.0 +5.2 | 43.8 +6.0 |
| **Image** | | | | | | | | | |
| Split 1 | 57.2 | 57.3 +0.2 | 56.7 -0.4 | 56.4 -0.8 | 53.4 -3.7 | 55.9 -1.3 | 57.3 +0.1 | 57.0 -0.1 | 56.0 -1.1 |
| Split 2 | 55.8 | 55.3 -0.4 | 55.8 +0.0 | 56.2 +0.4 | 56.0 +0.2 | 55.4 -0.4 | 54.5 -1.3 | 54.8 -1.0 | 56.3 +0.5 |
| Split 3 | 57.7 | 59.1 +1.4 | 58.7 +1.0 | 60.2 +2.5 | 57.0 -0.7 | 58.2 +0.6 | 64.1 +6.5 | 52.2 -5.5 | 58.4 +0.7 |
| *Average* | 56.9 | 57.2 +0.4 | 57.0 +0.2 | 57.6 +0.7 | 55.5 -1.4 | 56.5 -0.4 | 58.6 +1.8 | 54.7 -2.2 | 56.9 +0 |
| **WebNav** | 25.8 | 26.5 +0.7 | 26.3 +0.5 | 26.3 +0.5 | 26.0 +0.2 | 25.7 -0.1 | 26.5 +0.7 | 25.8 -0.1 | 26.3 +0.5 |

**Sentiment Analysis** Recent work shows LLMs overfit widely-used datasets due to data contamination (Balloccu et al., 2024; Sainz et al., 2023; Oren et al., 2023). To mitigate this, we evaluate on a domain-specific financial tweets sentiment dataset[3]. We create 32 synthetic class-balanced datasets (998 samples each) using eight prompt types: zero-shot, zero-shot with background knowledge, with train-time or test-time stock ticker info, and few-shot variants. We use `Qwen2.5-7B-Instruct`, `Qwen2.5-32B-Instruct`, `Llama3.1-8B-Instruct`, and `Llama3.3-70B-Instruct` (Grattafiori et al., 2024) for each prompt type. Background knowledge uses detailed guideline instructions for better task alignment. Stock tickers are sampled one at a time for synthesis. Details are in section C. We train task models using XGBoost and evaluate F1 score on a 2,388-item test set. Rubrics are compiled by randomly sampling 200 points from each real and synthetic dataset.

**Text2SQL** We evaluate SYNQUE on Text2SQL using three DBs from the BIRD benchmark (Movies, App Store, Computer Students — we the last two as Apps and Computers) (Li et al., 2023a). We synthesize 1,000 data points with 4 prompt types: zero-shot with background knowledge (guidelines and schema), zero-shot with test-time info (random table rows), and few-shot (three examples). `Qwen2.5-7B-Instruct` and `Llama3.1-8B-Instruct` models are used for dataset generation. Task models are finetuned from `Qwen2.5-Coder-1.5B-Instruct` following CodeS[4] and evaluated using execution accuracy on the real test set. For rubrics, we sample 30 points per synthetic and real dataset. Real data sizes are 60, 69, and 164 for Apps, Computers, and Movies respectively.

**Image Classification** In addition to text-only settings, we evaluate SYNQUE on image classification using synthetic datasets curated from `unmet-promise`[5]. These datasets are created with different prompts using Stable Diffusion 1.1 and 1.5: label, label plus physical relation, and label plus background description (Geng et al., 2025). Images are mapped to ImageNet classes (Deng et al., 2009) via caption analysis C, then filtered with a vision-language model to remove noisy labels. Data cleaning details are provided in Appendix section C. The final set includes 15 classes with 300 images each. Due to limited samples, the 15-class task is split into three 5-class tasks (table 4). We train ResNet-50 (He et al., 2015) from scratch for 50 epochs, early stopping on 10% validation data. Evaluation uses

Table 4: Division of the 15 selected ImageNet classes into three 5-class splits for image classification tasks. Each row corresponds to one split used in our experiments.

| Splits | ImageNet classes |
|---|---|
| 1 | bra, mask, lion, cloak, tank |
| 2 | hammer, backpack, stage, throne, tray |
| 3 | plate, desk, kimono, shield, church |

---

[3] https://huggingface.co/datasets/zeroshot/twitter-financial-news-sentiment
[4] https://github.com/RUCKBReasoning/codes
[5] https://huggingface.co/datasets/scottgeng00/unmet-promise

Table 3: Spearman (left) and Pearson (right) correlation scores of SYNQUE proxy metrics. LENS uses a fraction of samples for rubric compilation except for Web Navigation tasks.

| Tasks | LENS 7B debiased | biased | LENS 32B debiased | biased | PAD | MMD² | MDM | Mauve |
|---|---|---|---|---|---|---|---|---|
| **Sentiment** | .25 .33 | .26 .17 | .38 .26 | .24 .23 | .53 .65 | .45 .67 | .68 .85 | .53 .57 |
| **Text2SQL** | | | | | | | | |
| Computer | .19 .13 | .10 .10 | .41 .45 | .18 .23 | .46 .69 | .33 .85 | .39 .63 | .24 .78 |
| Apps | .38 .37 | .42 .49 | .46 .40 | .55 .61 | .43 .42 | .53 .79 | .44 .56 | .74 .52 |
| Movies | .41 .50 | .41 .26 | .50 .46 | .56 .47 | .50 .64 | .38 .46 | .61 .41 | .65 .68 |
| *Average* | .33 .33 | .31 .28 | .46 .43 | .43 .44 | .46 .58 | .41 .70 | .48 .53 | .55 .66 |
| **Image** | | | | | | | | |
| Split 1 | -.18 -.19 | -.30 -.35 | -.28 -.28 | -.68 -.67 | -.06 -.05 | .66 .52 | -.37 -.27 | -.04 -.20 |
| Split 2 | .02 -.04 | .14 .05 | .20 .05 | .20 .05 | .20 .31 | .09 .17 | .03 -.32 | .03 .13 |
| Split 3 | -.10 -.01 | -.15 -.03 | .31 .33 | .37 .44 | .02 -.15 | .26 .21 | -.54 -.76 | .46 .34 |
| *Average* | -.09 -.08 | -.10 -.11 | .08 .03 | -.04 -.06 | .05 .04 | .33 .30 | -.30 -.45 | .15 .09 |
| **WebNav** | .15 .17 | .11 .18 | .15 .15 | .08 .09 | .11 .08 | -.02 .06 | -.11 -.08 | -.09 -.10 |

mean reciprocal rank (MRR) for finer performance measurement. Rubrics are constructed from 100 sampled images per real and synthetic data, consistent across SYNQUE methods.

**Web Navigation**   Our fourth task evaluates SYNQUE on agentic web navigation planning using WebVoyager (He et al., 2024) and synthetic data from NNetNav (Murty et al., 2025). Inputs include task objectives, current step observations (accessibility tree), and past actions; targets are actions leading to task success. WebVoyager has 15 websites; we exclude Google Flights and Booking which are no longer feasible (Zhou et al., 2024; Murty et al., 2025), leaving 13 sites with 557 tasks. Each site forms a test domain split into 5 synthetic subsets. Models are fine-tuned with LoRA (Hu et al., 2021) on `Qwen2.5-7B-Instruct`. We use all synthetic and 20 real samples per method.

## 5.1 RESULTS ANALYSIS

**SYNQUE proxies correlate with task performance and improve selection.**   table 3 shows that SYNQUE proxy metrics demonstrate moderate to strong correlation with downstream task performance. To show the practical utility of these proxies, we simulate selecting the top 3 datasets based on each metric's score and compare their average task performance against the mean performance of all available synthetic datasets. As shown in table 2, nearly all proxy metrics significantly improve dataset selection over selecting synthetic datasets non-discriminately (i.e. uniform selection). This demonstrates that SYNQUE is an effective framework for maximizing real-data performance, despite not having access to labeled real data and only a limited sample of real data. We also conduct an experiment with PERPLEXITY on Text2SQL, to examine the effectiveness as a potential proxy. As shown in table 5, it correlates poorly with even text data, therefore we conclude that scaling methods would not work under our setting, where no annotated data is available for evaluation to build a regression model that predicts the best data mixture.

Table 5: Spearman (left) and Pearson (right) correlations with PERPLEXITY scoring on the BIRD Text2SQL benchmark

| | PERPLEXITY |
|---|---|
| Computers | -.31 -.33 |
| Apps | -.25 -.29 |
| Movies | .24 .31 |
| *Average* | -.11 -.10 |

**Performance on ambiguous image data shows high variance.**   The synthetic image classification data contains significant visual variability and label ambiguity, especially in Split 1 between classes like "stage" and "throne" (fig. 2a, fig. 2b). This confuses most proxy metrics, resulting in inconsistent correlations across splits (table 3). However, table 2 shows that when used for selection, several proxies (e.g. debiased LENS 32B, MMD²) still improve average task performance.

**Using more real samples improves correlation.**   As shown in table 6, increasing the number of unannotated real samples $m_r$ consistently leads to stronger correlations for all proxy metrics. This indicates that even a modest increase in available real-world data can significantly improve the reliability of synthetic data quality estimation.

Table 6: Spearman (left) and Pearson (right) correlations with different number of real samples for scoring Text2SQL. LENS uses debiased 32B scoring. Note that results in table 3 use 30 real samples.

| | $|\mathcal{U}_\mathrm{r}| = 25$ | | | | | $|\mathcal{U}_\mathrm{r}| = 50$ | | | | |
| | LENS | PAD | MMD² | MDM | Mauve | LENS | PAD | MMD² | MDM | Mauve |
|---|---|---|---|---|---|---|---|---|---|---|
| Computers | .22 .22 | .36 .65 | .28 .80 | .39 .65 | .19 .65 | .64 .38 | .51 .78 | .34 .87 | .39 .62 | .87 .78 |
| Apps | .29 .48 | -.20 -.23 | .40 .77 | .44 .56 | .65 .70 | .64 .66 | .68 .76 | .57 .80 | .44 .56 | .32 .65 |
| Movies | .33 .48 | .33 .36 | .52 .57 | .60 .40 | -.14 -.35 | .43 .49 | .57 .37 | .81 .56 | .67 .45 | .81 .60 |
| *Average* | .28 .39 | .16 .26 | .40 .71 | .47 .54 | .23 .33 | .57 .51 | .59 .64 | .57 .74 | .50 .55 | .67 .68 |

**LENS excels on complex, long-horizon tasks.** As shown in both tables, the 32B debiased LENS is the only proxy that consistently achieves positive correlation and improves top-3 task performance across all tasks and splits. Its advantage is particularly pronounced in web navigation, a complex planning task where representation-based metrics struggle. LENS leverages LLM reasoning over rich, structured inputs like accessibility trees to generate interpretable rubrics. For instance, an example characteristic point for the website "Wolfram Alpha" is: "Dataset B tasks focus on data retrieval (e.g. temperature anomalies, moon phases) while Dataset A emphasizes applied computational problem-solving". These specific nuances in long text (e.g. instructions, state observations) are difficult to capture using general-purpose dense vector embedders, which explains why LENS outperforms representation-based proxy metrics on complex, abstract tasks like web navigation. This method does exhibit weaker correlation in image classification. We hypothesize that this stems from the inability of VLMs to capture meaning characteristics descriptions in batches of images during rubric generation, and VLM rubric generation will likely improve as VLMs improve in quality.

## 5.2 ABLATION STUDIES AND COST ANALYSIS

**Cost-Effectiveness of SYNQUE Proxies** A key motivation for SYNQUE is efficiency. Our representation-based proxies require a one-time embedding computation, after which scoring all datasets is nearly instantaneous (e.g., 19 seconds for MMD on 32 datasets). LENS, using modern LLM serving frameworks, is also highly efficient, taking $\sim 15$ seconds per dataset with a 32B model on a H200 GPU. In contrast, perplexity-based data selection, inspired by scaling-law studies (Liu et al., 2025), require training many (e.g., 512 1M models used in their experiment) small models on the mixture of all synthetic datasets, a significantly more costly procedure, yet yield weaker correlations (table 3). This highlights the practical advantage of the SYNQUE framework.

**Larger scorers lead to stronger correlations** We find that larger scoring models yield stronger correlations between LENS and task performance, as shown in table 3. Intuitively, this is expected because larger models generally possess more robust instruction-following and reasoning capabilities, enabling them to better assess data quality and align proxy scores with downstream performance.

**LENS is robust to preferential bias in LLM training data.** To address concerns that an LLM evaluator might favor data it generated, we tested LENS with different scoring models on data generated by `Qwen2.5`. The results, detailed in Appendix table 7, show that performance is consistent across evaluators, including those distinct from models used to generate the synthetic data, indicating minimal preferential bias.

**Principled debiasing and rubrics are critical for LENS.** As illustrated in table 3, the correlations between LENS and task performance consistently increase when debiasing is applied, indicating that raw scores may be systematically biased and do not reliably reflect true data quality. Once debiasing is introduced, the correlation becomes strongly positive and consistently outperforms the biased scores. This demonstrates that debiasing effectively corrects for these systematic errors and aligns LENS scores with actual task performance. Further ablations show that using a rubric consistently improves correlation over a zero-shot baseline (Appendix table 9), and that 10 rubric points generally offers the best trade-off between specificity and generality (Appendix table 10).

## 6 CONCLUSIONS, LIMITATIONS AND FUTURE WORK

We formalized the SYNQUE problem of ranking synthetic datasets by their impact on real-world task performance using limited unannotated real data. Our comprehensive evaluation established that various proxies can reliably predict downstream performance, offering a cost-effective alternative to full model training. We proposed LENS, a novel proxy leveraging LLM reasoning and principled debiasing, which consistently outperforms others on challenging, long-horizon tasks. Overall, SYNQUE offers a robust framework for synthetic data selection when labeled real data is scarce.

SYNQUE assume that real data is scarce, a setting not all deployments face. While LENS performs well on the complex tasks studied, its effectiveness should be validated on more diverse tasks. Additionally, we experiment with limited-size LLMs due to resource constraints. Future work should explore 1) scaling LENS to larger sizes and different architectures, especially strong VLMs, to assess generality and improvements. 2) using rubric feedback to guide LLMs in synthesizing more realistic data, and 3) developing fine-grained, example-level proxy use to directly improve task model training.

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

## A   VISUALIZATION OF AMBIGUOUS IMAGE

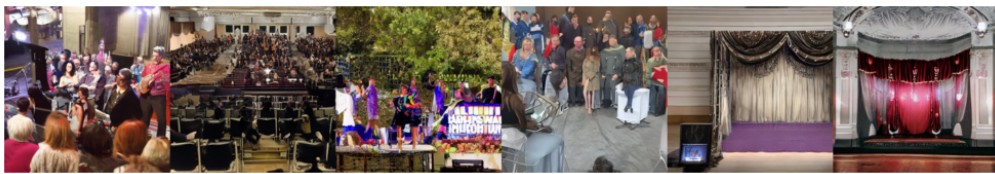

(a) Sample synthetic images of class "stage"

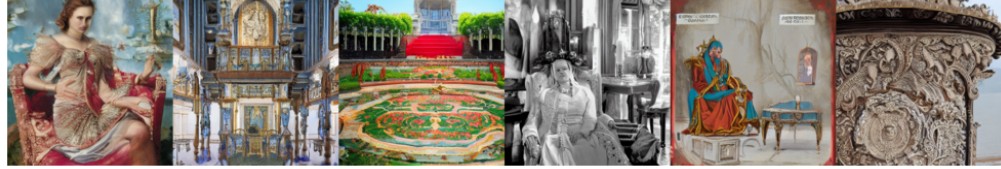

(b) Sample synthetic images of class "throne"

Figure 2: Visualization of synthetic images from second split for classes (a) "stage" and (b) "throne"

## B   ADDITIONAL ABLATIONS

### B.1   ABLATION STUDY ON LENS PREFERENTIAL BIAS

Table 7: We use two non-Qwen scoring models on tasks where synthetic data was generated by `Qwen2.5` models. The consistent positive correlations of the debiased scores demonstrate that LENS is robust to this potential bias.

| LENS (`Granite-8B`) | Debiased | | Biased | |
|---|---|---|---|---|
| Task | Spearman | Pearson | Spearman | Pearson |
| Sentiment | .21 | .30 | -.04 | -.02 |
| Text2SQL | .22 | .20 | .19 | .23 |
| LENS (`Ministral-8B`) | Debiased | | Biased | |
| Task | Spearman | Pearson | Spearman | Pearson |
| Sentiment | .28 | .39 | .52 | .31 |
| Text2SQL | .32 | .29 | .16 | .16 |

### B.2   LENS WITH DIFFERENT BACKBONE MODELS AND SIZES

As shown in table 8, LENS demonstrates robust ranking capabilities across a diverse range of backbone models and sizes. However, the impact of the debiasing step varies significantly by task complexity and model capability:

- **Text2SQL (Complex Generation):** Debiasing consistently improves or maintains correlation strength across nearly all models. For instance, with `Ministral-8B`, debiasing doubles the Spearman correlation (.16 → .32), and it yields the highest overall performance with `Qwen2.5-32B` (.46). This suggests that for complex structural tasks, removing distributional priors remains beneficial even for capable models.

- **Sentiment Analysis (Simple Classification):** The results are more nuanced. For models with severe initial bias, such as `Granite-8B-Instruct`, debiasing is critical, correcting a negative correlation (-.04) to a positive one (.21). However, for highly capable or larger models (e.g., `GPT-4.1`, `Granite-4.0 (32B)`, `Gemma3-27B`), debiasing tends to degrade performance. We hypothesize that these stronger models are already well-calibrated for simple classification tasks; consequently, our analytical debiasing term may introduce unnecessary noise that overrides the model's accurate intrinsic priors.

Table 8: Comparison of original and debiased LENS scores for ranking synthetic datasets across various backbone models and sizes. For GPT-4.1, sentiment analysis results are reported only due to cost constraints.

| LENS (Qwen2.5-7B-Instruct) | Debiased | | Original | |
|---|---|---|---|---|
| Task | Spearman | Pearson | Spearman | Pearson |
| Sentiment | .25 | .33 | .26 | .17 |
| Text2SQL | .33 | .33 | .31 | .28 |
| **LENS (Qwen2.5-32B-Instruct)** | Debiased | | Original | |
| Task | Spearman | Pearson | Spearman | Pearson |
| Sentiment | .38 | .26 | .24 | .23 |
| Text2SQL | .46 | .43 | .43 | .44 |
| **LENS (Ministral-8B-Instruct)** | Debiased | | Original | |
| Task | Spearman | Pearson | Spearman | Pearson |
| Sentiment | .28 | .39 | .52 | .31 |
| Text2SQL | .32 | .29 | .16 | .16 |
| **LENS (Granite-8B-Instruct)** | Debiased | | Original | |
| Task | Spearman | Pearson | Spearman | Pearson |
| Sentiment | .21 | .30 | -.04 | -.02 |
| Text2SQL | .22 | .20 | .19 | .23 |
| **LENS (Granite-4.0-h-small) (32B)** | Debiased | | Original | |
| Task | Spearman | Pearson | Spearman | Pearson |
| Sentiment | .16 | .39 | .50 | .44 |
| Text2SQL | .32 | .36 | .40 | .48 |
| **LENS (Gemma3-12B-Instruct)** | Debiased | | Original | |
| Task | Spearman | Pearson | Spearman | Pearson |
| Sentiment | .10 | .13 | .30 | .18 |
| Text2SQL | .30 | .34 | .25 | .25 |
| **LENS (Gemma3-27B-Instruct)** | Debiased | | Original | |
| Task | Spearman | Pearson | Spearman | Pearson |
| Sentiment | .17 | .33 | .37 | .31 |
| Text2SQL | .31 | .38 | .31 | .27 |
| **LENS (GPT-4.1-2025-04-14)** | Debiased | | Original | |
| Task | Spearman | Pearson | Spearman | Pearson |
| Sentiment | .07 | .30 | .28 | .37 |

### B.3 ABLATION STUDY ON LENS RUBRIC DESIGN

Table 9: LENS with vs. without a rubric on sentiment analysis, showing the rubric's effectiveness. We use `Qwen2.5-7B-Instruct` for scoring.

| LENS Model | w/ Rubric | Spearman | Pearson |
|---|---|---|---|
| Qwen2.5-7B-Instruct | NO | 0.23 | 0.21 |
| Qwen2.5-32B-Instruct | NO | 0.32 | 0.13 |
| Qwen2.5-7B-Instruct | YES | 0.25 | 0.33 |
| Qwen2.5-32B-Instruct | YES | 0.38 | 0.26 |

Our ablation results from table 9 demonstrate the effectiveness of the rubric component of LENS. Table 10 demonstrates that generating too few or too many points will degrade the performance of LENS.

### B.4 ABLATION STUDY ON $MMD^2$ KERNEL FUNCTIONS

Based on the ablation results from table 11, the Laplacian kernel performs best on text tasks, but all kernels show limited effectiveness on more complex domains.

Table 10: Bottom: Varying the number of rubric points, where 10 points provides a good balance.

| # Rubric Points | Sentiment Analysis | | Text2SQL (Avg) | |
|---|---|---|---|---|
| (Debiased) | Spearman | Pearson | Spearman | Pearson |
| 5 | 0.14 | 0.19 | 0.38 | 0.30 |
| 10 | 0.25 | 0.33 | 0.33 | 0.33 |
| 15 | -0.06 | 0.05 | 0.23 | 0.21 |

Table 11: Pearson / Spearman correlation coefficients of $\text{MMD}^2$ with different kernel functions.

| Task | Polynomial | RBF | Laplacian | Linear | Sigmoid |
|---|---|---|---|---|---|
| Sentiment | .67 / .45 | .67 / .45 | **.78 / .60** | .67 / .45 | .67 / .46 |
| Text2SQL (avg) | .51 / .43 | .52 / .44 | **.70 / .51** | .52 / .44 | .53 / .44 |
| Image (avg) | .22 / .30 | .22 / .30 | .21 / .30 | .22 / .30 | .22 / .30 |
| WebNav | .06 / -.02 | .06 / -.01 | .05 / -.04 | .06 / -.02 | .05 / -.03 |

## B.5 ABLATION STUDY ON MDM WITH DIFFERENT NUMBER OF CLUSTERS

Table 12: Pearson / Spearman coefficients of **MDM** with different number of clusters (K=3, 5, 10, 20) on sentiment analysis.

| K=3 | K=5 | K=10 | K=20 |
|---|---|---|---|
| .85/ .68 | .86/ .68 | .86/ .69 | .86/ .69 |

The table with varying number of medoid clusters for **MDM** shows that changing the number of clusters has negligible effect on the correlation between **MDM** and F1 score.

## B.6 ABLATION STUDY ON USING DIFFERENT ENCODER FOR REPRESENTATION-BASED METRICS

The results with BGE-M3[6] demonstrate that representation-based metrics are measuring fundamental distribution shifts rather than artifacts of a specific encoder.

## B.7 CORRELATION OF PAD PREDICTIONS FROM DIFFERENT CLASSIFIERS VS. F1 SCORES

In summary, according to table 14, classifier choice plays a critical role in **PAD** prediction performance, with the random forest model demonstrating the highest correlation with F1 scores. This may be due to the ensemble nature of both the random forest and the sentiment analysis classifier (XGBoost), possibly enhancing alignment between predictions and F1 outcomes, while the MLP shows weaker correspondence.

## B.8 TRADE-OFF ANALYSIS BETWEEN LENS AND DIVERSITY MEASURE

To further understand the interplay between LENS and the diversity measure, we conducted an experiment using a "hybrid" score, defined as $\alpha \cdot \text{LENS} + (1 - \alpha) \cdot \text{MDM}$. As shown in fig. 3, our analysis reveals that increasing $\alpha$thereby placing greater weight on LENS and less on diversityresults in a monotonic decrease in both the Pearson and Spearman correlation coefficients. This trend suggests that the LENS and diversity measures do not demonstrate synergistic effects when combined.

## B.9 ABLATION ON THE NUMBER OF SAMPLES USED FOR SCORING ON SIMPLE TASK

We investigate the sensitivity of LENS proxies to the size of the unannotated real dataset $|\mathcal{U}_r|$ for the sentiment analysis task.As shown in table 15, representation-based metrics (**PAD**, $\text{MMD}^2$,

---

[6]https://huggingface.co/BAAI/bge-m3

Table 13: Pearson / Spearman correlation coefficients of proxies on sentiment analysis with BGE-M3 as the encoder.

| MAUVE | PAD | MMD$^2$ | MDM |
|---|---|---|---|
| .60/ .65 | .75/ .65 | .82/ .67 | .86/ .74 |

Table 14: Correlation Coefficients (Pearson / Spearman) of **PAD** predictions from different classifiers vs. F1 scores.

| Classifier | Correlation Coefficients (P/S) |
|---|---|
| Logistic Regression | .65 / .53 |
| Random Forest | .75 / .64 |
| 2-layer MLP | .56 / .43 |

Table 15: Pearson / Spearman correlation between the LENS score and the test F1 score for different metrics with differet number of real instances used for scoring (100 and 200).

| $|\mathcal{U}_r| = 100$ | | | | | $|\mathcal{U}_r| = 200$ | | | | |
|---|---|---|---|---|---|---|---|---|---|
| LENS | PAD | MMD$^2$ | MDM | MAUVE | LENS | PAD | MMD$^2$ | MDM | MAUVE |
| -.03/-.13 | .65/.47 | .68/.44 | .85/.70 | .54/.51 | .26/.38 | .65/.53 | .67/.45 | .85/.68 | .57/.53 |

MDM, MAUVE) demonstrate remarkable robustness in this domain, maintaining strong correlations (e.g., **MDM** Spearman 0.70) even with as few as 100 samples. This aligns with our hypothesis that embedding spaces effectively capture distributional shifts in simpler, stylistic tasks. In contrast, LENS requires a larger sample size to stabilize for this specific task 5, exhibiting weak correlations at $|\mathcal{U}_r| = 100$ (Pearson $-0.03$, Spearman $-0.13$) before recovering significantly to a positive Spearman correlation of 0.38 at $|\mathcal{U}_r| = 200$. This behavior highlights a distinct complexity trade-off: while representation metrics are highly data-efficient for stylistic domains, they fail on complex, long-horizon tasks such as Web Navigation. Conversely, LENS proves robust in those more complex domains even with very limited unlabled real data.

## C  ADDITIONAL EXPERIMENTAL DETAILS

### C.1  DATA CURATION

**Sentiment analysis**  For sentiment analysis, we used a cleaned version of the original validation split as the test set, removing URLs from the data. Synthetic samples are generated and validated to ensure each output is non-empty, alphabetic 'headline' and a 'sentiment' label restricted to the values '0', '1', or '2'. Only samples meeting these structural and content criteria were retained for downstream analysis.

**Text2SQL**  During Text2SQL data synthesis, we validate generated question–SQL pairs by ensuring both fields are non-empty, contain alphanumeric content, and are formatted as a dictionary with 'question' and 'SQL' keys. For non-zero-shot generations, SQL queries are executed against the target database; pairs failing execution are discarded. This process enforces correct structure, meaningful content, and SQL executability.

**Image classification**  We classify each image in the `unmet-promise` dataset into one of three categories: *label_only*, *label_relation*, or *label_background*, based on its caption. Captions are lowercased and stemmed. If a class-specific background keyword appears in the caption, the image is assigned to *label_background*. If a relation keyword is present, it is assigned to *label_relation*. Images not matching either are assigned to *label_only*. We balance the number of samples per category and class, and store the processed data for downstream tasks. To further ensure fidelity of images, we use `Qwen2.5-VL-7B-Instruct` to filter noisy images. Prompt used for filtering is provided in listing 8.

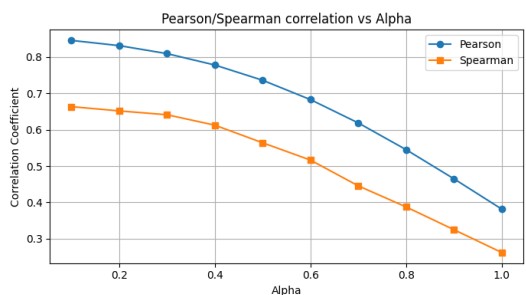

Figure 3: Pearson/Spearman correlation coefficients of the hybrid score vs. alpha. Both correlation coefficients are monotonically decreasing as alpha increases (more LENS impact and less MDM impact).

**Web navigation** We first group individual trajectories into tasks. Then we use different seeds to create 5 disjoint subsets with equal amount of tasks. We fine-tune using LoRA rank of 64 and the open-instruct [7] fine-tuning codebase.

## C.2 SYNQUE SCORING

For **PAD**, we reserve 20% of all embeddings as a holdout test set to train the classifier, and compute the classification error on this set to obtain the final **PAD** score. For $\text{MMD}^2$, we generate the proxy score using a polynomial kernel [8] with degree 3 and Coef0 parameter 1. For **MDM**, we use Fasterpam (Schubert & Rousseeuw, 2019) to compute $k$ medoids in the embeddings, setting $k$ to 3 for sentiment analysis and 5 for other tasks, using Euclidean distance for clustering; **MDM** is then calculated by averaging the Euclidean distance from each data point to its corresponding medoid. For LENS, the prompts used for rubric compilation and scoring are provided in section C.5.3 and section C.5.4, respectively. For MAUVE calculation, we use the default hyperparameter, the same embedding model as the other representation-based proxies (*qte-Qwen2-7B-Instruct*), and the scaling factor is 5. For PERPLEXITY calculation in Text2SQL, we first reformat each Text2SQL question as a string: `"question": <sample>`. We then use `Qwen2.5-7B` to compute the perplexity, considering only the tokens corresponding to the question text (i.e., the <sample> portion), and excluding the prompt tokens (such as `"question":`).

## C.3 STANDARD DEVIATION OF CORRELATION COEFFICIENTS

## C.4 EXAMPLE RUBRICS

Listing 3: Example characteristic descriptions $C_{\text{s,r}}$

```
"Dataset B consistently specifies the analyst behind actions...",
"Dataset B maintains strict financial focus without political or
    entertainment tangents present in A...",
"Dataset B entries always directly connect stock movements to specific
    analyst actions...",
"Dataset B shows more frequent price target amount disclosures...",
"Dataset B uses standardized financial terminology consistently...",
"Dataset B maintains neutral tone in earnings reports...",
"Dataset B focuses exclusively on institutional analyst
    perspectives...",
"Dataset B headlines strictly follow '[Analyst] [Action] on [Ticker]
    [Rationale]' structure...",
"Dataset B contains no social media tags/hashtags...",
"Dataset B shows higher frequency of ETF coverage..."
```

---

[7]https://github.com/allenai/open-instruct

[8]https://scikit-learn.org/stable/modules/generated/sklearn.metrics.pairwise.polynomial_kernel.html

Table 16: Standard deviation of Spearman (left) and Pearson (right) correlation coefficients of SYNQUE proxy metrics across 5 seeds. **MDM** only scores on synthetic datasets therefore no change in input data.

| Tasks | LENS 7B | | LENS 32B | | PAD | MMD | MDM |
|---|---|---|---|---|---|---|---|
| | debiased | biased | debiased | biased | | | |
| **Sentiment** | .23 .17 | .12 .14 | .09 .11 | .07 .07 | .02 .02 | .02 .02 | .00 .00 |
| **Text2SQL** | | | | | | | |
| Computer | .48 .39 | .38 .44 | .21 .21 | .18 .14 | .04 .08 | .03 .08 | .00 .00 |
| Apps | .38 .34 | .25 .21 | .22 .22 | .22 .14 | .36 .48 | .05 .15 | .00 .00 |
| Movies | .32 .32 | .33 .24 | .22 .24 | .11 .14 | .07 .13 | .05 .14 | .00 .00 |
| *Average* | .39 .35 | .32 .30 | .22 .22 | .17 .14 | .15 .23 | .04 .12 | .00 .00 |
| **Image** | | | | | | | |
| Split 1 | .47 .48 | .51 .51 | .39 .38 | .28 .22 | .08 .06 | .00 .02 | .00 .00 |
| Split 2 | .43 .47 | .43 .42 | .40 .41 | .43 .53 | .11 .08 | .00 .04 | .00 .00 |
| Split 3 | .52 .39 | .47 .39 | .40 .42 | .29 .22 | .07 .07 | .00 .05 | .00 .00 |
| Average | .47 .44 | .47 .44 | .40 .40 | .33 .32 | .09 .07 | .00 .04 | .00 .00 |
| **WebNav** | .15 .17 | .11 .13 | .09 .11 | .15 .19 | .04 .02 | .02 .02 | .00 .00 |

Listing 4: Example characteristic descriptions $C_{r,s}$

```
"Dataset B includes headlines without stock tickers ...",
"Dataset B contains non-financial news ...",
"Dataset B incorporates social media-style commentary...",
"Dataset B includes international/non-English company names...",
"Dataset B references non-institutional analysts/sources...",
"Dataset B features headlines about dividends...",
"Dataset B includes legal/regulatory actions unrelated to markets...",
"Dataset B uses technical trading jargon...",
"Dataset B contains macroeconomic commentary without stock links...",
"Dataset B includes non-company-specific index/currency forecasts..."
```

## C.5 LLM USAGE

**Model serving**  We use vLLM (Kwon et al., 2023) to serve open source models such as for data synthesis and dataset scoring. For `Llama3.3-70B-Instruct` model, we use Ollama [9] Q4_K quantized version to construct synthetic datasets for sentiment analysis. We use 2 * Nvidia A40 48GB GPUs for other models for synthesis and scoring. To improve LLMs' generation throughput, we use vLLM's batched inference feature and enable prefix-caching to further improve generation efficiency.

**LLM hyperparameter**  For LENS rubric compilation and scoring, we set temperature to 0 and top_p to 0.95.

### C.5.1 DATA SYNTHESIS PROMPTS

Listing 5: Zero-shot prompt used for sentiment analysis dataset generation.

```
Generate three realistic financial news headlines for sentiment analysis.

Guidelines for Generating Headlines:

Sentiment Labeling:
Each headline must be assigned a sentiment label based on its tone:
    -    Bearish (0): Indicates negative sentiment about a stock or market
         trend.
    -    Bullish (1): Indicates positive sentiment about a stock or market
         trend.
    -    Neutral (2): Indicates neutral or informational tone.

Now, generate three new financial news headlines following these
    guidelines. Please use JSON format and generate one type of each
    sentiment label (0, 1, 2) in your response.
```

---

[9] https://ollama.com/

Listing 6: Zero-shot with background knowledge prompt used for sentiment analysis dataset generation.

```
Generate three realistic financial news headlines about stock tickers
    following real-world financial reporting for sentiment analysis.

Guidelines for Generating Headlines:

1. Format & Style:
    -   Headlines must be concise and mimic real financial news.
    -   Use sentence case formatting (capitalize only the first word and
        proper nouns).
    -   Some headlines should start with a stock ticker (e.g., $AAPL -),
        while others should begin with the company name or a broader
        market trend.

2. Ticker Inclusion:
    -   At least one headline should include a stock ticker (e.g., $TSLA
        - or $NVDA -).
    -   Some headlines should refer to companies by name instead of
        tickers (e.g., "Alphabet and Meta see price targets cut at
        Barclays").

3. Common Financial Themes:

Ensure headlines reflect realistic financial news topics,
including:
    -   Stock downgrades/upgrades
    -   Price target adjustments
    -   Market trends/economic outlook
    -   Company performance concerns
    -   Company news
    -   Company announcements
    -   Company events

4. Source Attribution:
    -   When relevant, mention an investment firm, analyst, or research
        group (e.g., Morgan Stanley, Barclays, Oppenheimer).
    -   Do not fabricate research firms-use only well-known institutions.

5. Sentiment Labeling:

Each headline must be assigned a sentiment label based on its tone:
    -   Bearish (0): Indicates negative sentiment about a stock or market
         trend.
    -   Bullish (1): Indicates positive sentiment about a stock or market
         trend.
    -   Neutral (2): Indicates neutral or informational tone.

Sentiment Labeling:
Each headline must be assigned a sentiment label based on its tone:
    - Bearish (0): Indicates negative sentiment about a stock or market
        trend.
    - Bullish (1): Indicates positive sentiment about a stock or market
        trend.
    - Neutral (2): Indicates neutral or informational tone.

Now, generate three new financial news headlines following these
    guidelines. Please use JSON format and generate one type of each
    sentiment label (0, 1, 2) in your response.
```

Listing 7: Zero-shot with background and stock ticker information prompt used for sentiment analysis dataset generation.

```
Generate three realistic financial news headlines about stock tickers
    following real-world financial reporting for sentiment analysis.

Guidelines for Generating Headlines:

1. Format & Style:
    -   Headlines must be concise and mimic real financial news.
```

```
        -    Use sentence case formatting (capitalize only the first word and
             proper nouns).
        -    Some headlines should start with a stock ticker (e.g., $AAPL -),
             while others should begin with the company name or a broader
             market trend.

2. Ticker Inclusion:
        -    At least one headline should include a stock ticker (e.g., $TSLA
             - or $NVDA -).
        -    Some headlines should refer to companies by name instead of
             tickers (e.g., "Alphabet and Meta see price targets cut at
             Barclays").

3. Common Financial Themes:

Ensure headlines reflect realistic financial news topics, including:
        -    Stock downgrades/upgrades
        -    Price target adjustments
        -    Market trends/economic outlook
        -    Company performance concerns
        -    Company news
        -    Company announcements
        -    Company events

4. Source Attribution:
        -    When relevant, mention an investment firm, analyst, or research
             group (e.g., Morgan Stanley, Barclays, Oppenheimer).
        -    Do not fabricate research firms-use only well-known institutions.

5. Sentiment Labeling:

Each headline must be assigned a sentiment label based on its tone:
        -    Bearish (0): Indicates negative sentiment about a stock or market
              trend.
        -    Bullish (1): Indicates positive sentiment about a stock or market
              trend.
        -    Neutral (2): Indicates neutral or informational tone.

Now, generate three new financial news headlines about stock tickers: {
    stock_ticker} following these guidelines. Please use JSON format and
    generate one type of each sentiment label (0, 1, 2) for diversity.
```

### C.5.2 DATA CLEANING PROMPTS

Listing 8: Prompt used for filtering noisy synthetic images

```
You are a helpful assistant that filters out an image. You will be given
    an image and its corresponding text caption.

You should return true if the primary object in the image is not a ${
    label} in common sense. Return false otherwise.

Image:
{image}

Caption:
{caption}
```

### C.5.3 LENS RUBRIC COMPILATION PROMPTS

Listing 9: Rubric compilation prompt used in sentiment analysis (commonalities)

```
You are a world class data analyst on financial news headline. You will
    be given some financial news headline samples from dataset A and
    dataset B. Based on the provided similar characteristics between them
    , list how B is similar to A. Return {num} points as a JSON list of
    strings. Please focus on specific and granular similarities between
    the two datasets, your generated characteristic points should apply
    to all the samples from the two datasets.

Samples from A:
{A}

Samples from B:
{B}
```

Listing 10: Rubric compilation prompt used in sentiment analysis (differences)

```
You are a world class data analyst on financial news headlines. You will
    be given some financial news headline samples from dataset A and
    dataset B. Based on the provided similar characteristics between them
    , list how B is {feedback} A. Please focus on granular differences
    between the two datasets, your generated characteristic points should
     apply to all the samples from the corresponding dataset (A or B).
    Return {num} points as a JSON list of strings.

Similar characteristics between A and B:
{similar_points}

Samples from A:
{A}

Samples from B:
{B}
```

Listing 11: Rubric compilation prompt used in Text2SQL (commonalities)

```
You are a world class data analyst on database queries in natural
    language. Given samples from dataset A and dataset B, list how B is {
    feedback} A. Return {num} points as a JSON list of strings. Please
    focus on specific and granular similarities between the two datasets,
     your generated characteristic points should apply to all the samples
    .

Question samples from A:
{A}

Question samples from B:
{B}
```

Listing 12: Rubric compilation prompt used in Text2SQL (differences)

```
You are a world class data analyst on database queries in natural
    language. Given query samples from dataset A and dataset B. Based on
    the provided similar characteristics between them, list how B is {
    feedback} A. Return {num} points as a JSON list of strings. Please
    focus on specific and granular differences between the two datasets,
    your generated characteristic points should apply to all the samples
    from the corresponding dataset (A or B).

Similar characteristics between A and B:
{similar_points}

Question samples from A:
{A}

Question samples from B:
{B}
```

Listing 13: Rubric compilation prompt used in image classification (commonalities)

```
Below are {num_image} images from dataset B:
{B}

Below are {num_images} images from dataset A:
{A}

Given some samples from image classification dataset A and dataset B,
    list how dataset B is similar to dataset A. Return ${num_points}
    points that summarize the similar characteristics of the two datasets
    . Focus on the characteristics of the image in terms of how they are
    structured, styled, or captured (e.g., lighting, background,
    composition, etc.) rather than the image specifications such as
    resolution, size, etc. Your generated characteristic points should
    apply to all the samples from the corresponding dataset (A or B).
    Output should be a JSON list of strings.

Your listed points:
```

Listing 14: Rubric compilation prompt used in image classification (differences)

```
Below are {num_image} images from dataset B:
{B}

Below are {num_images} images from dataset A:
{A}

Given some samples from image classification dataset A and dataset B,
    list how dataset B is different from dataset A. Similar
    characteristics are provided below for reference. Return ${num_points
    } points that summarize the characteristics of the two datasets (e.g
    ., dataset A is ... dataset B is ...). Focus on the characteristics
    of the images in terms of how they are structured, styled, or
    captured (e.g., lighting, background, composition, etc.) rather than
    the image specifications such as resolution, size, etc. Your
    generated characteristic points should apply to all the samples from
    the corresponding dataset (A or B). Output should be a JSON list of
    strings.

Similar characteristics:
${similar_characteristics}

Your listed points:
```

Listing 15: Rubric compilation prompt used in web navigation (commonalities)

```
Given some samples of web navigation tasks and the web accessibility tree
    of dataset A and dataset B, list how B is {feedback} A. Return {num}
    points that summarize the characteristics of the two datasets. The
    two accessibility trees generated from the same website are provided.
```

```
      Please only list out characteristics that are related to the
   proposed web navigation tasks, the accessibility trees are provided
   to only help you understand the context of the proposed tasks,
   therefore do not mention the accessibility tree in your response.
   Please focus on granular and specific characteristics, and your
   generated characteristic points should apply to all the samples.
   Output should be a JSON list of strings.

Accessibility Tree for dataset A:
{A_tree}

Sampled web navigation tasks from dataset A:
{A}

Accessibility Tree for dataset B:
{B_tree}

Sampled web navigation tasks from dataset B:
{B}
```

Listing 16: Rubric compilation prompt used in web navigation (differences)

```
Given some samples of proposed web navigation tasks and the web
   accessibility tree of dataset A and dataset B. Based on the similar
   characteristics between them. List how B is {feedback} A. Return {num
   } points that summarize the characteristics of the two datasets. The
   two accessibility trees of the same website are provided. Please only
    list out characteristics that are related to the proposed web
   navigation tasks, the accessibility trees are only provided to help
   you understand the context of the proposed tasks. Therefore do not
   list any characteristics that are related to the accessibility tree
   in your response. Please focus on granular and specific
   characteristics, and your generated characteristic points should
   apply to all samples in corresponding dataset (A or B). Output should
    be a JSON list of strings.

Similar characteristics between A and B:
{similar_points}

Accessibility Tree for dataset A:
{A_tree}

Sampled web navigation tasks from dataset A:
{A}

Accessibility Tree for dataset B:
{B_tree}

Sampled web navigation tasks from dataset B:
{B}
```

### C.5.4 LENS SCORING PROMPTS

Listing 17: Scorer prompt used in sentiment analysis

```
You are given similarities and differences between two financial news
   headline datasets A and B.

Your task is to judge how likely is the given financial news headline
   comes from dataset {prediction}. Answer your judgement with one of
   the following strings: "very unlikely", "unlikely", "unsure", "likely
   ", and "very likely".

Similar characteristics between dataset A and B:
{similar_characteristics}

Differences between dataset A and B:
{differences}

Financial news headline sample to be judged:
```

```
{example}

Your judgement in JSON format:
```

Listing 18: Scorer prompt used in Text2SQL

```
You are given similarities and differences between datasets A and B about
    database queries in natural language.

Your task is to judge how likely is the given database query in natural
    language comes from dataset {prediction}. Answer your judgement with
    one of the following strings: "very unlikely", "unlikely", "unsure",
    "likely", and "very likely".

Similar characteristics between dataset A and B:
{similar_characteristics}

Differences between dataset A and B:
{differences}

Natural language database query to be judged:
{example}

Your judgement in JSON format:
```

Listing 19: Scorer prompt used in image classification

```
You are given similarities and differences between datasets A and B.

Your task is to judge how likely is the given image comes from dataset {
    prediction}. Answer your judgement with one of the following strings:
     "very unlikely", "unlikely", "unsure", "likely", and "very likely".

Format:
{format_instructions}

Similar characteristics between dataset A and B:
{similar_characteristics}

Differences between dataset A and B:
{differences}

Image to be judged:
{image}

your judgement in JSON format:
```

Listing 20: Scorer prompt used in web navigation

```
You are given similar and different characteristics between two datasets
    A and B consisting of web navigation tasks.

Your objective is to judge how likely is the given web browsing task
    comes from dataset {prediction}. Answer your judgement with one of
    the following strings: "very unlikely", "unlikely", "unsure", "likely
    ", and "very likely".

Format:
{format_instructions}

Similar characteristics between dataset A and B:
{similar_characteristics}

Different characteristics between dataset A and B:
{differences}
```

```
Web navigation task to be judged:
{example}

Your judgement in JSON format:
```

