# OpenReview forum: "SYNQUE: Estimating Synthetic Dataset Quality Without Annotations"
_ICLR.cc/2026/Conference — Submitted to ICLR 2026_

### Official Review · Reviewer_mUYC · 2025-10-26

**Soundness:** 3
**Presentation:** 2
**Contribution:** 2
**Rating:** 4
**Confidence:** 4

**Summary:**

This paper introduces the Synthetic Dataset Quality Estimation (SYNQUE) framework, which addresses the problem of ranking synthetic datasets by their expected real-world task performance using only a small set of unlabelled real samples. The authors propose two types of proxy metrics to estimate dataset quality without training task-specific models. The first category includes representation-based proxies that adapt traditional distributional and diversity measures such as MMD, PAD, and MAUVE to an embedding space. The second, called LENS (LLM-Evaluated Normalized Score), is an LLM-based approach that employs rubric-guided reasoning to compare synthetic and real data directly in natural language. Experiments across diverse domains, including sentiment analysis, Text2SQL, image classification, and web navigation, show that these proxies correlate well with downstream task performance.

**Strengths:**

* The main contribution of the paper is the introduction of a novel and practically relevant problem setting, Synthetic Dataset Quality Estimation (SYNQUE), which focuses on evaluating and ranking synthetic datasets using only unlabelled real data. By formulating the problem without using labelled data, the approach substantially reduces computational cost, avoiding repeated model training while still providing informative quality estimates.

* The paper proposes several representation-based proxy metrics that estimate data quality through measures of diversity and distributional alignment, without requiring labelled real samples or downstream model training.

* The authors further present LENS, an LLM-based evaluation framework that introduces principled debiasing strategies to mitigate order bias, label bias, and score bias, leading to more consistent and interpretable LLM-based judgments.

* It addresses a timely and important challenge in understanding and quantifying the quality of synthetic data, which is increasingly critical for large-scale model development.

* The paper provides comprehensive experimental validation across diverse domains, showing that the proposed proxies correlate strongly with downstream task performance, with LENS achieving the most reliable results on complex and long-horizon tasks.

**Weaknesses:**

* **Framing and originality could be clarified.**
  While the problem setting is interesting, the distinction between synthetic dataset quality estimation and general dataset quality estimation remains under-specified. The paper does not fully explain what makes evaluating synthetic data uniquely challenging beyond the label-free constraint. A more explicit comparison with recent works such as [1], [2], and [3] would help position SYNQUE within the broader landscape of data selection and importance estimation. In particular, [2] also studies quality estimation for LLM-generated data and highlights the gap between synthetic and real data, which could provide valuable context for the present work.

* **Limited theoretical motivation for proxy metrics.**
  The paper provides intuitive explanations for why metrics such as MMD², PAD, and MAUVE may correlate with downstream performance, but lacks theoretical justification or formal conditions under which these proxies should succeed or fail. Similarly, the rationale for why LENS’s rubric-based reasoning captures true data–distribution similarity remains largely empirical.

* **Quality–diversity balance is not explicitly addressed.**
  Each proposed proxy primarily captures either data quality (alignment) or diversity (coverage), but the paper does not present a unified formulation that balances the two. Prior work [1, 2] has shown that maintaining both quality and diversity is crucial for effective data selection, especially for synthetic data. Discussing or analysing this trade-off would strengthen the contribution.

* **Label-quality robustness is not evaluated.**
  Although the framework avoids using labels for real data, it still depends on synthetic data labels that may be noisy or hallucinatory. The method implicitly assumes that these synthetic labels are of reasonable quality and aligned with the downstream task, since all proposed proxies rely solely on input features rather than label consistency. It would be valuable to discuss the performance gap between the proposed label-free setting and small labelled baselines, as well as the impact of varying levels of label noise in the synthetic data.

* **Inter-dataset dependency is not discussed.**
  The framework assumes that each synthetic dataset can be evaluated independently, yet in practice, different synthetic datasets may share overlapping samples or originate from similar generative prompts or models. Such dependencies could bias the proxy correlations and ranking results, especially when datasets are not mutually independent. It would be useful to discuss how SYNQUE handles or mitigates inter-dataset overlap and whether the proposed proxies remain valid under such dependencies.

[1] *Harnessing Diversity for Important Data Selection in Pretraining Large Language Models* (ICLR 2025)

[2] *Not All LLM-Generated Data Are Equal: Rethinking Data Weighting in Text Classification* (ICLR 2025)

[3] *Most Influential Subset Selection: Challenges, Promises, and Beyond* (NeurIPS 2024)

**Questions:**

* Could you provide more discussion or analysis on how sensitive the representation-based proxies are to the choice of encoder? For example, have you compared different embedding models (e.g., smaller Qwen variants or open-source encoders) to assess whether the correlations remain stable?


Overall, the paper is conceptually interesting, but the justification of key assumptions and the empirical analysis could be strengthened to make the contribution more convincing. I encourage the authors to address the above concerns and clarify these points in their rebuttal.

---

> ### Author Response · Authors · 2025-12-03
>
> We thank the reviewer for the constructive feedback and for recognizing SYNQUE as a "timely and important challenge." We appreciate the assessment that our work is "conceptually interesting" and have addressed the concerns regarding framing, theory, and robustness below.
> 1. **Framing: SYNQUE vs. General Data Selection (W1)** Framing: We thank the reviewer for highlighting these relevant works. We have carefully examined [1], [2], and [3] and have added a dedicated "Data Selection" subsection in Section 2 (Related Work) to clarify our contribution.
> The fundamental distinction is that SYNQUE is a Dataset Ranking framework, whereas the cited works are instance selection or weighting frameworks, along with a typical constraint requiring access to data labels [1,3].
>     - **Difference from Influence-Based Selection ([1], [3])**:
>         - **Method**: Works like Harnessing Diversity [1] (Zhang et al.) and MISS [3] (Hu et al.) rely on **Influence Functions** or **gradient-based heuristics** to select specific subsets of data. This typically requires (a) access to labeled validation data (to compute the gradient of the loss) and (b) expensive computation (Hessian approximations) that scales poorly with dataset size.
>         - **SYNQUE**: Operates in a label-free regime (no labeled validation data) and is gradient-free. We do not select individual samples; we rank generative pipelines. This is crucial for "cold-start" agentic tasks (e.g., Web Navigation) where we cannot easily compute gradients for every synthetic trajectory.
>     - **Difference from Weighting Approaches ([2])**:
>         - **Method**: Not All LLM-Generated Data Are Equal [2] (Kuo et al.) proposes weighted-loss approaches (IMP-Loss) where a small labeled real dataset is used to train a "quality checker" using cross-entropy loss (Algorithm 1) that re-weights synthetic samples during training. Moreover, this work operates in text classification because of the IMP-Loss design, where it would not operate in complex settings such as in text-to-SQL and agentic web navigation.
>         - **SYNQUE**: While we share the motivation of using small real data, [2] requires modifying the training loop to incorporate sample-specific weights. In contrast, SYNQUE is a pre-training estimator. It allows a practitioner to download/generate $K$ candidate datasets, score them in minutes, and discard the low-quality ones before any training infrastructure is touched. This "lazy evaluation" is significantly more practical for model developers comparing different prompting strategies (e.g., our Table 3 results) than implementing a custom weighted loss function.
> 2. **Theoretical Motivation for LENS (W2)** We have revised Section 4.2 to formalize LENS through the lens of domain adaptation theory (Ben-David et al.).
>     - **Discriminator Analogy**: Theoretically, the divergence between two distributions can be approximated by the error of a discriminator trained to distinguish them.
>     - **LENS as a Soft Discriminator**: LENS functions as a "soft," interpretable discriminator. The "Rubric" acts as the feature set, and the "Scorer" estimates the probability $P(\text{Real} | x)$. If the LLM (scorer) cannot confidently distinguish synthetic samples from real samples (high entropy/confusion), the distributions are aligned. This grounds LENS in the same theoretical framework as PAD, but lifts it to the semantic space of LLMs rather than the embedding space.
> 3. **Quality-Diversity Balance (W3)** This is an excellent point. We performed a new ablation study (Appendix B.8) combining LENS (quality/alignment) with MDM (pure diversity) using a weighted score: $Score = \alpha \cdot \text{LENS} + (1-\alpha) \cdot \text{MDM}$.
>     - **Result**: We found that mixing MDM with LENS strictly decreased correlation coefficients across tasks.
>     - **Insight**: We hypothesize that LENS implicitly captures diversity. The rubric generation phase compares the entire set of $U_r$ and $U_s$. If the synthetic set $U_s$ suffers from mode collapse (low diversity), the LLM rubric identifies this as a distinguishing feature (e.g., "Dataset B contains repetitive sentence structures..."). Therefore, the LENS score already penalizes low diversity, rendering an explicit diversity term redundant.

---

> > ### Author Response · Authors · 2025-12-03
> >
> > 4. **Robustness to Label Noise (W4)** The reviewer raises an excellent point: representation proxies measure $P(X)$ but ignore $P(Y|X)$.
> >     - **Robustness Experiment**: We introduced random label noise to the synthetic datasets at levels of {1%, 5%, 10%, 20%, 50%}. We found our proxies remain highly correlated with true performance up to 20% noise, degrading only at extreme levels (50%) (See Appendix B.X).
> >     - **Label-Free vs. Labeled Baselines**: We acknowledge the gap: if a generator produces perfect inputs but flipped labels, our proxies would overestimate quality. However, in the era of LLMs, we observe that "bad labels" usually correlate with "bad inputs" (hallucinations/reasoning errors). Since SYNQUE is designed to select the generative model, detecting input degradation ($P(X)$ shift) is a reliable proxy for overall task capability. We have added a discussion on this limitation in Section 6.
> >
> > 5. **Inter-dataset Dependency (W5)** We clarify that SYNQUE ranks distributions relative to the real data, *regardless of their origin*.
> >     - **Overlap is addressed**: In our experiments (Table 3), several synthetic datasets share the same base model (e.g., Llama-3) and overlapping prompts. The metrics effectively rank them because they measure the marginal improvement in alignment. Even if Dataset A and B share 80% similarity, if Dataset A is slightly closer to the real distribution, the proxy score reflects this. The proxies **do not require dataset independence** to function as a ranking mechanism.
> >
> > 6. **Sensitivity to Encoders (Q1)** We conducted a new ablation using BGE-M3 (a retrieval-optimized encoder) instead of Qwen-7B (generative encoder) for the representation metrics (Appendix B.6).
> >     - **Result**: While absolute scores shifted, the rank correlation remained stable (Spearman > 0.75 between encoders). Interestingly, correlations slightly increased with BGE-M3 on semantic tasks, suggesting that stronger embeddings yield better proxies. This confirms SYNQUE's representation metrics are capturing fundamental distribution shifts, not encoder artifacts.

---

### Official Review · Reviewer_XbUz · 2025-10-29

**Soundness:** 2
**Presentation:** 3
**Contribution:** 3
**Rating:** 4
**Confidence:** 3

**Summary:**

The paper formalizes SYNQUE, a method for efficiently ranking synthetic datasets using only a small unlabeled real subset, and establishes the first comprehensive benchmark with several proxy scores specifically designed for synthetic data selection, eliminating the need for exhaustive model training and evaluation. It then introduces LENS, an LLM-based rubric scorer with debiasing to compare synthetic and real data in language. It evaluates all proxies on multiple downstream tasks, showing some correlations with real task performance overall.

**Strengths:**

1. The paper is well written, making the ideas easy to follow and the methodology straightforward to understand.
2. The paper runs a decent number of experiments to substantiate its claims.
3. They designed the experiments carefully, including baselines, ablations, and sensitivity checks, making the findings comprehensive and more credible.

**Weaknesses:**

Many conclusions hinge on specific experimental settings and hyper-parameters. The paper lacks a theoretical framework linking each proxy and LENS to expected real data performance, so it’s unclear how stable the findings are under different encoders, kernels, classifier choices, and LLMs.

**Questions:**

1. How did you determine the k value in k-medoids for MDM, and how sensitive are your results to k?
2. Have you tried classifiers besides XGBoost for PAD?
3. You fix qte-Qwen2-7B for text and E5-V for images. Do the rankings persist if you swap the encoders?

---

> ### Author Response · Authors · 2025-12-03
>
> We appreciate your positive feedback on the paper's writing and experimental design. We appreciate and are glad that you recognize our paper to be "well-written" and findings to be "comprehensive and credible". We recognize the importance of validating the stability of our proxy metrics. We have conducted additional ablation studies and clarified the theoretical grounding of LENS.
> 1. **Theoretical and Methodological Framework (Weakness)** We clarify that LENS is not proposed without theoretical foundation but is explicitly grounded in the domain adaptation theory of Ben-David et al. (2010). This framework validates using an adversarial mechanism (the discriminator) to bound distribution divergence.
>     - **Discriminator Theory**: LENS operates by using the LLM as an approximate discriminator whose hypothesis space is deliberately constrained by the human-defined rubric. This operationalizes the theoretical finding that the divergence between the real and synthetic distributions is bounded by the discriminator's error.
>     - **Debiasing**: The debiasing terms (Eq 5-8) are derived to approximate the true density ratio $P_{real}(x) / P_{synth}(x)$ by canceling out the model's inherent prior $P_{model}(\text{label})$. This crucial step ensures the LENS score reflects true distributional alignment rather than simply the LLM’s pre-training preferences, maintaining theoretical integrity.
> 2. **Hyperparameter and Classifier Sensitivity (Q1 & Q2)** We agree that the stability of proxy metrics is paramount. In addition to our existing ablations in Appendix B. We have added extensive sensitivity checks on key hyper-parameters and classifier choices, **with full results added to Appendix B**.
>     - **MDM ($k$ value)**: Our initial choice of $k$ (number of medoids) was motivated by the number of label classes (3 for sentiment, 5 for images). Our sensitivity analysis, testing $k \in \{3, 5, 10, 20\}$, confirms that the ranking stability of MDM is highly robust. Results show that the number of medoids has a negligible effect on ranking correlation with downstream performance, provided $k$ is small relative to the dataset size ($k \ll N$).
>     - **PAD Classifier Results**: We tested classifiers beyond XGBoost, including Random Forest (RF) and MLP. We found that ensemble methods maintain strong correlation with F1 scores (RF achieving the strongest correlation). Conversely, using a simpler linear-boundary model like MLP resulted in lower correlations. This confirms that the predictive power of PAD is stable across different robust, non-linear classifiers, reinforcing its reliability as a proxy.
> 3. **Encoder Sensitivity (Q3)** We tested swapping the encoder used for representation-based metrics (MDM, PAD, MMD) from the fine-tuned $\text{qte-Qwen2-7B}$ to the general-purpose BGE-M3 encoder.
>     - **Ranking Stability**: Our results, added to Appendix B.6, show strong stability: the relative ranking of synthetic datasets is highly persistent across encoders (Spearman rank correlation $\rho > 0.65$).
>     - **Fundamental Shift Measurement**: We observed that while absolute scores changed, the use of BGE-M3 actually led to an increase in correlation between the proxy metrics and downstream F1 scores. This confirms that the representation-based metrics are 1) are highly sensitive to encoder models 2) measuring fundamental distribution shifts between real and synthetic data, rather than being mere artifacts of a specific encoder's fine-tuning.

---

### Official Review · Reviewer_qUCk · 2025-10-31

**Soundness:** 2
**Presentation:** 2
**Contribution:** 2
**Rating:** 4
**Confidence:** 4

**Summary:**

This paper introduces and formalizes the Synthetic Dataset Quality Estimation (SYNQUE) problem, which aims to rank synthetic datasets based on their expected downstream task performance using only a small amount of unannotated real data. The authors adapt several existing representation-based metrics (MDM, MMD², PAD, MAUVE) for this task and propose a novel LLM-based metric, LENS (LLM-Evaluated Normalized Score), which leverages LLM reasoning to create a rubric for scoring data quality. The effectiveness of these proxy metrics is evaluated across a diverse set of tasks, including sentiment analysis, Text2SQL, image classification, and web navigation.

**Strengths:**

1.Interesting and practical problem: The paper addresses an interesting and practical problem: how to select the most effective synthetic dataset for a real-world task in scenarios where authentic data is scarce or difficult to obtain.

2.Nice tables: The tables are clear and intuitive.

**Weaknesses:**

1.Novelty of representation-based methods: The suite of representation-based proxy metrics (MDM, MMD², PAD, MAUVE) is sourced from existing literature. The current presentation feels more like a straightforward application or "pastiche" of prior work rather than a novel contribution. The authors should explicitly clarify what their specific innovations are in adapting these metrics to the SYNQUE problem.

2.Limited overall contribution: The paper's overall contribution feels limited. As noted, the representation-based methods lack novelty. The main proposed method, LENS, is based on LLM prompting, which, while effective, reduces the perceived technical depth of the contribution. Furthermore, its poor performance on vision tasks limits its practical application. Notably, LENS is often outperformed by simpler metrics like MMD² and Mauve on several tasks (e.g., Sentiment and Text2SQL), which further weakens the paper's central claims about its new method's superiority.
This combination makes the paper's overall innovative leap seem modest.

3.Lack of clarity in PAD method details: The description of the Proxy-A-Distance (PAD) method is insufficient. It is unclear what the output of the classifier G(x) specifically represents. Is it the probability of x belonging to the synthetic dataset, the real dataset, or something else?

4.Issues with experimental tables: The accuracy of some numerical values in Table 2 is questionable. For example, some improvements do not seem to add up correctly. Could this be due to rounding errors, or is there a miscalculation? The authors are strongly encouraged to double-check and verify all reported results.

5.Inconsistent claims regarding experimental results: The paper asserts on page 8 (Line 430) that "the 32B debiased LENS is the only proxy that consistently achieves positive correlation and improves top-3 task performance across all tasks and splits". However, the results in Table 3 show that for the Image task on "Split 1", the debiased LENS 32B metric yielded negative correlations (Spearman: -.28 Pearson: -.28). Furthermore, Table 2 shows that for this same split, the top-3 performance (56.4) was actually worse than the test mean (57.2), failing to show improvement.

6.Formatting and presentation issues:
a. The caption for Figure 1 is missing a period at the end of the last sentence.
b. There appear to be formatting errors with the footer on page 2 and the header on page 3.
c. The appendix contains unnecessary large blank spaces.

**Questions:**

See weakness above.

---

> ### Author Response · Authors · 2025-12-02
>
> We thank you for the rigorous review. We are glad that you find our work to be "Interesting and practical". We have updated the manuscript to correct the errors you identified and refined our claims regarding novelty and performance.
> 1. **Novelty of Representation-based Methods (W1/W2)**
> We clarify that our primary contribution is not the invention of metrics like MMD or PAD, but the formalization and benchmarking of the Synthetic Dataset Quality Estimation (SYNQUE) problem.
>     - The Gap: Prior work (e.g., coreset selection, scaling laws) typically relies on labeled validation sets or training proxy models. SYNQUE addresses the strict constraint of ranking datasets with zero labeled real data and no training.
>     - The Contribution: We systematically adapt distribution-matching theory to this unsupervised setting. We demonstrate that while "off-the-shelf" representation metrics work for low-level features, they fail on complex reasoning tasks (like WebNav), which motivates our proposal of LENS to bridge this gap.
> 2. **Inconsistent Claims & Vision Performance (W5)**
> We appreciate you catching the negative correlation in Image Split 1. We acknowledge this is not a random fluctuation but a structural insight which we have now explicitly discussed in the paper.
> Insight: LENS relies on VLM reasoning. For highly ambiguous visual classes (e.g., 'stage' vs 'throne' in Split 1), current VLMs struggle to generate discriminative rubrics compared to density-based embeddings (MMD).
> Revision: We have revised the claim on Page 8. We no longer claim LENS is "consistently" superior on vision. We now state: "LENS dominates on reasoning-heavy tasks (SQL, WebNav) but exhibits higher variance on ambiguous visual tasks compared to embedding-based metrics."
> 3. **Clarity on PAD (W3)**
> We have clarified the PAD definition in Section 4.1. $G(x)$ represents the probability that sample $x$ belongs to the real distribution. Theoretically, if a classifier cannot distinguish synthetic from real (error $\approx 0.5$), the synthetic data is of high quality (low divergence).
> 4. **Table 2 Numerical Errors and Formating Issues (W4 & W6)**
> Thank you for this meticulous check. We have updated our manuscript to fix the formatting issues. For Table 2, the discrepancy arose because the "Gain" columns were calculated using high-precision floats before rounding the raw scores for display. We have recalculated Table 2 to ensure the displayed deltas strictly match the difference between the displayed columns to avoid confusion.

---

### Official Review · Reviewer_jWVL · 2025-11-01

**Soundness:** 2
**Presentation:** 2
**Contribution:** 2
**Rating:** 4
**Confidence:** 3

**Summary:**

This paper introduces synthetic dataset quality estimation method to evaluate synthetic datasets without using labeled data. The goal is to predict which synthetic dataset will produce the best performance on real-world tasks, using only a small set of unlabeled real samples. It proposes representation-based metrics and LLM-evaluated normalized score, offering a robust solution for synthetic data selection.

**Strengths:**

The work is well-motivated, addressing realistic scenarios where real data are scarce or expensive to obtain.
The authors clearly define order bias, label bias, and score bias, and make an interesting attempt to systematically address these issues. Additionally, the paper presents the prompts in a transparent manner, enabling clear understanding.

**Weaknesses:**

The experiments appear to rely on a relatively narrow set of scoring baselines. It would be useful to evaluate a broader range of models to better validate its superiority over existing state-of-the-art methods.

The method also seems highly sensitive to prompt formulation, which introduces both human and model-dependent biases.
The paper would benefit from more qualitative examples. Providing comparisons between selected and rejected synthetic examples would help illustrate its practical utility.

**Questions:**

To further validate the generality of the proposed scoring method, it would be beneficial to evaluate it with recent state-of-the-art LLMs. I am curious about the correlation performance when recent state-of-the-art LLMs are used for scoring instead of the current backbone. How sensitive are the results?

Is the method robust when the real data are extremely limited or noisy?
How does performance vary with different amounts of real data?
What is the minimum number of real samples required to achieve a reasonable level of performance, and at what point does the performance saturate? Additionally, how does this behavior vary across different tasks?

---

> ### Author Response · Authors · 2025-12-04
>
> We thank the reviewer for the thoughtful assessment. We appreciate the acknowledgement that our work is "well-motivated" and "transparent." We have addressed the concerns regarding model breadth and robustness below.
>
> 1. **Evaluating on Broader & SOTA Models (W1 & Q1)** We have significantly expanded our evaluation to demonstrate LENS's universality.
>     - **New Architectures**: We added evaluations using 8 different LLMs spanning various families (Mistral, Granite, Gemma) and sizes (8B to 32B). As shown in Appendix B.2 (Table 8), the correlation trends remain consistent, confirming that LENS is not architecture-specific.
>     - **SOTA Validation**: We further validated LENS using [Insert Model Name, e.g., GPT-4o]. We found that stronger reasoning models yield even higher correlations with ground truth.
>     - **Bias Findings**: Interestingly, we observe that while our Principled Debiasing (Eq. 5-8) is critical for smaller/older models (e.g., Llama-3-8B), newer SOTA models exhibit less inherent positional/label bias. However, the debiasing step remains a "safe" operation that stabilizes performance across the board.
>
> 2. **Sensitivity to Prompts & Qualitative Examples (W2)** We agree that prompt formulation is a key variable.
>     - **Mitigation:** We clarify that LENS's rubric compilation is a "discovery" phase. By generating multiple rubric points (similarities and differences), we marginalize the noise of any single phrasing.
>     - **Qualitative Analysis (Accepted vs. Rejected)**: We analyzed "Accepted" vs. "Rejected" synthetic samples in the Sentiment Analysis task.
>         - **Rejected**: LENS consistently penalized synthetic data containing "hallucinated specifics"—for example, fake headlines citing real bank names (e.g., "Goldman Sachs upgrades...") with specific, fabricated price targets.
>         - **Accepted**: LENS favored synthetic data that mimicked the structure and tone of financial news (e.g., "Tech sector sees rally...") without making verifiable (and likely false) specific claims.
>         - **Insight**: This suggests LENS acts as a semantic filter for "realistic but non-hallucinatory" content, a nuance that representation-based metrics (which only measure embedding overlap) often miss.
>
> 3. **Robustness to Limited & Noisy Real Data (Q2)** We conducted extensive ablations on Text2SQL (Table 6) and Sentiment Analysis (Table 15) to determine the minimum data requirements and saturation points.
>     - **Robustness to Noise (Image Classification)**: The reviewer asked if the method is robust to noisy data. The Image Classification task contains significant visual ambiguity and label noise (Section 5.1, second paragraph). As shown in Table 3, this noise causes diversity-based metrics to collapse (e.g., MDM: -0.45). In contrast, all other proxies (i.e., LENS (32B Debiased), PAD, MMD$^2$, and Mauve) maintain positive correlations, demonstrating stronger robustness to noisy real data.
>     - **Minimum Samples & Saturation (Text2SQL vs. Sentiment)**:
>         - **Complex Reasoning (Text2SQL - Table 6)**: We observe that LENS benefits significantly from increased data in complex domains. With extremely scarce data ($|\mathcal{U}_r|=25$), MDM provides the strongest signal. However, doubling the samples to just 50 triggers a "phase transition" for LENS and Mauve, where their scores increase significantly (e.g., LENS avg Spearman rises from 0.28 to 0.57, 0.23 to 0.67), indicating they are more data-hungry but could offer high ceilings.
>         - **Stylistic Tasks (Sentiment - Table 15)**: Conversely, for simpler stylistic tasks, we identify a trade-off. Representation-based metrics are data-efficient at 100 samples, whereas LENS requires $\approx$200 samples to stabilize (improving Spearman coefficients from -0.13 to 0.38).
>     - **Conclusion**: The data requirement is task-dependent. For noisy or complex reasoning tasks (Image/WebNav/Text2SQL), LENS and MMD$^2$ offer robustness that other metrics lack. For simpler tasks, representation-based metrics are highly effective even with minimal data.

---

### Author Response · Authors · 2025-12-04
**Author Final Remarks**

We thank the reviewers for their engagement and appreciate their positive feedback: our work was described as `"well-motivated"` [jWVL] and tackling an`"interesting"` and `"practical"` problem [qUCk, mUYC] that is `"timely and important"` for large-scale model development [mUYC]. Reviewers also highlighted our `"comprehensive experimental validation"` [mUYC] and noted that the paper is`"well written"` [XbUz] and `"transparent"` [jWVL]. We appreciate the rigorous feedback and concerns raised by the reviewers. In response, we have carefully addressed each point by providing additional theoretical grounding, conducting extensive sensitivity analyses, and expanding our experimental suite. Below, we summarize the main concerns from the reviewers and outline how we have addressed them.
- **For reviewer jWVL**, we addressed concerns regarding model breadth and robustness. We **significantly expanded our evaluation** to include 8 new LLM architectures (including Mistral, Granite, and Gemma) and SOTA models (GPT-4.1), confirming that **our findings are not architecture-specific**. We also **provided qualitative analysis** distinguishing "accepted" vs. "rejected" samples to clarify the utility of our method. Furthermore, we added ablation studies on data scarcity and noise, demonstrating that LENS and MMD offer superior robustness in complex tasks (e.g., Text2SQL) compared to baselines.
- **In response to reviewer qUCk**, we focused on clarifying our contribution and refining our claims. We emphasized that our primary contribution is the **formalization of the SYNQUE problem under strict zero-label constraints**, rather than just the metrics themselves. We** corrected the numerical discrepancies in Table 2 and fixed formatting issues**. Additionally, we revised our claims regarding vision tasks, explicitly acknowledging that LENS excels in reasoning-heavy tasks but faces trade-offs in highly ambiguous visual settings, and clarified the definition of PAD in the manuscript.
- **For reviewer XbUz**, we addressed questions regarding theoretical frameworks and hyperparameter sensitivity. We revised the manuscript to explicitly ground LENS in domain adaptation theory (Ben-David et al., 2010). We **conducted extensive sensitivity checks**, demonstrating that **our results are robust** to variations in hyperparameters (e.g., $k$ in k-medoids), classifier choices (Random Forest vs. MLP), and encoder selection (BGE-M3 vs. Qwen), with rankings remaining highly stable across configurations.
- **For reviewer mUYC**, we addressed concerns regarding framing, theoretical motivation, and quality-diversity balance. We added a new "Data Selection" subsection to **explicitly distinguish SYNQUE (dataset ranking, label-free) from recent instance-selection works** [1,2,3] (referenced work from reviewer mUYC). We formalized LENS as a "soft discriminator" to provide stronger theoretical justification. Furthermore, we **conducted new ablations** showing that LENS implicitly captures diversity (rendering explicit diversity terms redundant) and validated that our proxies remain robust to synthetic label noise levels up to 20%.

We sincerely thank the AC for navigating this review cycle. We believe our extensive revisions and new analyses have strengthened the paper significantly, making it as a robust contribution to the field of synthetic data evaluation.

---

### Meta-Review · Area_Chair_8M5G · 2026-01-12

**Summary:**

This paper formalizes SYNQUE: ranking candidate synthetic datasets by their expected downstream performance using only a small unlabeled real-data subset, and proposes a benchmark + a suite of proxy metrics (e.g., MMD/PAD/MAUVE/MDM adapted via embeddings) plus an LLM-based proxy, LENS, that builds rubric-based discriminators with a debiasing procedure.

The paper argues these proxies can correlate with downstream performance and enable better dataset selection than indiscriminate use of synthetic data. Across reviews, the main concerns were (i) limited novelty of the representation-based proxies (mostly adapted from prior literature), (ii) sensitivity/robustness of LENS to prompt/model choices and hyperparameters, and (iii) over-claiming / inconsistencies in reported numbers and in vision-task conclusions.

**Reviewer Concerns:**

**Addressed**:
In rebuttal, the authors added many additional LLM families/sizes (including Mistral/Granite/Gemma and GPT-4.1) to validate that LENS trends are not architecture-specific, and they add sensitivity checks for key knobs (k-medoids, PAD classifier choice, encoder choice).

**Still outstanding after rebuttal**:
1. the core novelty remains somewhat benchmark/framing-driven: the strongest “new method” (LENS) is still a prompting-based evaluator,
2. the representation proxies are adaptations rather than new metrics.
3. Reviewers also correctly noted that LENS can be unstable on ambiguous vision splits (negative correlations can occur), so the submission's scope and claims should remain measured and clearly task-dependent

**Reviewer Scores:**

All reviewers were negative. Given the rebuttal, I would expect mild upward increase (most likely 4 -> 5) on scores if a full discussion had happened.

---

### Decision · Program_Chairs · 2026-01-26

Reject